# Respiratory kinematics and the regulation of subglottic pressure for phonation of pitch jumps – a dynamic MRI study

Louisa Traser[1,2,3]*, Fabian Burk[4], Ali Caglar Özen[3,5,6], Michael Burdumy[5], Michael Bock[3,5], Daniela Blaser[2], Bernhard Richter[1,3], Matthias Echternach[7]

1 Institute of Musicians' Medicine, Medical Center–University of Freiburg, Freiburg, Germany, 2 Division of Phoniatrics, Department of Otorhinolaryngology, Head Neck Surgery, Inselspital, Bern University Hospital, University of Bern, Bern, Switzerland, 3 Faculty of Medicine, University of Freiburg, Freiburg, Germany, 4 Department of Otorhinolaryngology, Head- and Neck Surgery, Christian-Albrechts-University, Kiel, Germany, 5 Department of Radiology, Medical Physics, Medical Center–University of Freiburg, Faculty of Medicine, University of Freiburg, Freiburg, Germany, 6 German Consortium for Translational Cancer Research Partner Site Freiburg, German Cancer Research Center (DKFZ), Heidelberg, Germany, 7 Division of Phoniatrics and Pediatric Audiology, Department of Otorhinolaryngology, Munich University Hospital, Munich, Germany

* Louisa.Traser@uniklinik-freiburg.de

**Data Availability Statement:** All relevant data are within the paper and its Supporting Information files.

## Abstract

The respiratory system is a central part of voice production as it contributes to the generation of subglottic pressure, which has an impact on voice parameters including fundamental frequency and sound pressure level. Both parameters need to be adjusted precisely during complex phonation tasks such as singing. In particular, the underlying functions of the diaphragm and rib cage in relation to the phonation of pitch jumps are not yet understood in detail. This study aims to analyse respiratory movements during phonation of pitch jumps using dynamic MRI of the lungs. Dynamic images of the breathing apparatus of 7 professional singers were acquired in the supine position during phonation of upwards and downwards pitch jumps in a high, medium, and low range of the singer's tessitura. Distances between characteristic anatomical landmarks in the lung were measured from the series of images obtained. During sustained phonation, the diaphragm elevates, and the rib cage is lowered in a monotonic manner. During downward pitch jumps the diaphragm suddenly changed its movement direction and presented with a short inspiratory activation which was predominant in the posterior part and was associated with a shift of the cupola in an anterior direction. The magnitude of this inspiratory movement was greater for jumps that started at higher compared to lower fundamental frequency. In contrast, expiratory movement of the rib cage and anterior diaphragm were simultaneous and continued constantly during the jump. The data underline the theory of a regulation of subglottic pressure via a sudden diaphragm contraction during phonation of pitch jumps downwards, while the rib cage is not involved in short term adaptations. This strengthens the idea of a differentiated control of rib cage and diaphragm as different functional units during singing phonation.

**Funding:** The authors received no specific funding for this work.

**Competing interests:** The authors have declared that no competing interests exist.

## 1. Introduction

Key parameters which are regulated during human voice production include sound pressure level (SPL), fundamental frequency ($f_o$) and harmonic richness, and are particularly important in singing. The effector units which control these parameters are the vocal fold oscillations, the vocal tract (VT) and the breathing apparatus. While for regulation of SPL and harmonic richness all three effector units play a major role, $f_o$ is mainly controlled by the breathing apparatus and vocal fold stiffness: The active and passive forces of the breathing apparatus on the closed vocal folds create the subglottic pressure ($p_{sub}$). The increase of $p_{sub}$ leads directly to an increase of $f_o$ (and its related subjective pitch) [1]. Additionally, $f_o$ correlates with the oscillating mass of the vocal folds [2]—stiffer vocal folds and a higher driving pressure ($p_{sub}$) are required for a higher $f_o$ [3]. Thus, control of $p_{sub}$ is essential for singing in tune, and, in turn, careful adaptation of $p_{sub}$ is needed for singing different pitches. This relationship is shown in Fig 1. The question of how the breathing apparatus regulates $p_{sub}$ during phonation is a focus of voice pedagogy, voice therapy and voice research, however the details are still not understood.

As the diaphragm (DPH) is an inspiratory muscle it has been assigned a subordinated role for phonation in early studies [4]. Later investigations of transdiaphragmatic pressure during phonation noticed that professional singers activated the DPH for pitch jumps, probably with the goal of reducing $p_{sub}$ [5], or for fine regulation when the abdominal wall muscles are forcefully contracting (e.g., at the end of phonation) [6, 7]. Research on the regulatory role of the breathing apparatus during phonation in the past used techniques such as bodyplethysmography [8], transdiaphragmmatic pressure measurements [5, 8–10], magnetometry [11, 12], and respitrace [12–14]. However, these techniques only detect the impact of the respiratory system on the body's surface, or they measure solely cumulative effects of DPH and rib cage (RC) movement. They are limited concerning a differentiated analysis of movement of different

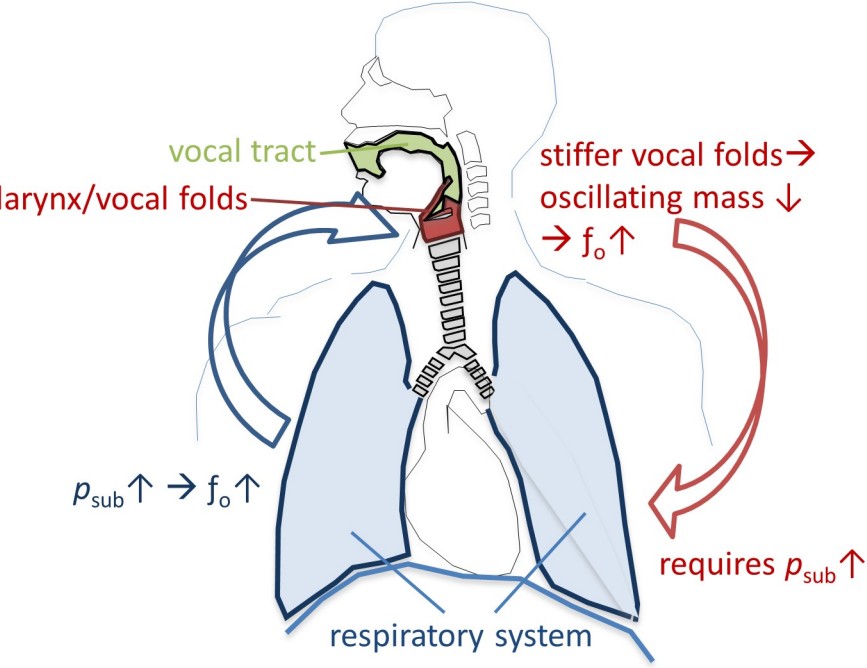

**Fig 1. Relationship between fundamental frequency ($f_o$) and subglottic pressure ($p_{sub}$) for phonation of different pitches.**

parts of the respiratory system during phonation. To the author's knowledge only one attempt has been made using an imaging technique (ultrasound) during phonation. In this pilot study Pettersen et al. visualized the DPH movement in three voice students by tracking the anterior DPH in the right hypochondrium and the posterior part indirectly via the left kidney movement via ultrasound [15]. They described differences in the movement pattern of the dorsal DPH section (bumpy movement patterns for ascending pitch) while the anterior sections moved in close to linear patterns. They also found, that during phonation the majority of singers moved the anterior section of the DPH considerably more than they moved their dorsal section. This stands in contrast to DPH imaging studies during respiration [16, 17] and a pilot study of our working group during phonation [18] which showed that the magnitude of DPH movement was twice as strong in the back, compared to the anterior part. Using ultrasound the comparability of the different measured distances seems questionable, considering that the measurement of the posterior part can only be acquired indirectly (via kidney movement).

Recent improvements to imaging hardware and software have led to the possibility of dynamic imaging of different parts of the respiratory apparatus simultaneously [16, 19–23] using dynamic magnetic resonance imaging (MRI). In a pilot study [18], the movements of the breathing apparatus of 6 professional singers during sustained phonation with a constant $f_o$ and SPL were analysed. Here, feasibility of simultaneous, dynamic DPH and RC imaging during phonation using MRI was shown and a very sophisticated movement pattern was observed: While the posterior and medial section of the DPH elevated quickly in the beginning of subjects' maximum phonation time, the anterior section elevated and RC descended slower. The opposite occurred with these movement velocities at the end of phonation, not dependent on pitch or loudness conditions.

But, the pilot study only investigated respiratory movements during sustained phonation. Singing is seldom reduced to phonation of a single pitch but characterized by pitch jumps of different magnitude and direction. Still, how different parts of the breathing system move with sudden pitch changes, is not understood in detail. It is of interest as in voice pedagogy a huge variety of instruction can be found of how the respiratory system should be used for best results in pitch jump phonation [24] and a mal regulation of phonatory breathing is believed to be associated with voice disorders [25]. Thus, the aim of this study was to evaluate the movements of the breathing apparatus during phonation of pitch jumps in professional singers.

The following hypotheses were formulated (see also Table 1): (A) As $p_{sub}$ influences pitch, it is expected that $p_{sub}$ is required to increase for upward jumps, and decrease for downward jumps. It was therefore hypothesised that respiratory movements differ with jump direction in regards to movement direction. (B) For sustained phonation, different movement patterns have been observed for different parts of the respiratory system [18]. Therefore, the second hypothesis was that the regulation of pitch jumps would not affect all parts of the respiratory system equally. In our pilot study [18], the movement range of the posterior DPH was twice as

**Table 1. Description of hypothesis A-D.**

| Nr. | hypotheses |
|-----|------------|
| A | Respiratory movements of the breathing apparatus differ with jump direction in regards to movement direction. |
| B | Movements of the respiratory system during pitch jumps are greater in the back part of the DPH compared to the anterior part of DPH or the rib cage. |
| C | More pronounced movements of the breathing apparatus occur during pitch jumps in the higher $f_o$ range compared to the lower $f_o$ range (high-medium vs. medium-low jumps). |
| D | More pronounced movements of the breathing apparatus occur when the same jump is performed on high vs. low lung volume (early vs. late in the task). |

large as the anterior DPH during sustained phonation. Thus, we additionally hypothesized that there would be differences in $p_{sub}$-adaptive movements of the anterior compared to posterior DPH during pitch jumps. (C) Greater $f_o$ changes can be obtained by lung pressure at shorter vocal fold lengths because the amplitude-to-length ratio is greatest when the vocal folds are short and lax [26]. Therefore, it was postulated that lager movements could be detected when pitch jumps were performed in higher $f_o$ range compared to lower $f_o$ range. (D) Due to the constant changes in recoil forces during expiration it was also hypothesised that a jump earlier in phonation, i.e. a pitch jump on higher lung volume, would lead to more pronounced movements of the breathing apparatus compared to a pitch jump later in phonation (on low lung volume).

## 2. Methods

### 2.1 Subjects and tasks

This study was approved by the Medical Ethics Committee of the University of Freiburg (Nr.273/14). 7 singers, professionally trained in western classical singing, took part. Professional singers were chosen as subjects, because it can be assumed that these subjects are, through education and training, less distracted by the noise during the MR imaging as they are used to auditory masking (e.g. in choir singing). They also use a very consistent and economic breathing strategy [8, 12, 14, 15, 27–29]. Table 2 shows the subjects'age, gender, voice classification, classification according to the Bunch and Chapman taxonomy [30] (a classification of professionalism) and relevant physical characteristics (vital capacity = VC, forced expiratory volume in one second = FEV1, height, weight). VC and FEV1 were obtained in a clinical setup using a ZAN100 spirometer (ZAN, Oberthulba, Germany) according to [31]. At the time of the recording, none of the participants complained of any vocal complaints, history of voice disorders, or respiratory pathologies (which was confirmed by the VC and FEV1 values in Table 2).

The phonation tasks were chosen according to the voice classification of the singer and represent a low (L), medium (M) and high pitch (H) in the tessitura of the respective repertoire of the singer (see Fig 2 for musical notes and corresponding $f_o$). The subjects were asked to phonate sustained-pitch notes with a rapid change to a higher or lower octave in a line of pitch jumps from high-to-medium-to-low-to-medium-to-high pitch with no pitch repetitions or breaths between each jump (= HMLMH, later referred to as *task 1*) as well as in reverse order (LMHML, later referred to as *task 2*). The subjects were asked to phonate in their western classically trained voice without a given register specification as they would on stage. Only satisfying recordings (for both the singer and the investigators) were included in the analysation and

**Table 2. Subject number, age, gender, voice classification, classification according to the Bunch and Chapman [30] taxonomy, vital capacity (= VC), forced expiratory volume in one second (= FEV1), body height and weight.**

| Subject | Age | Gender | Voice Classifi-cation | Bunch/ Chapman taxonomy | VC in l | FEV1 in l/s | Height in cm | Weight in kg |
|---|---|---|---|---|---|---|---|---|
| 1 | 28 | Female | Soprano | 4.5 | 4.72 | 3.88 | 165 | 65 |
| 2 | 25 | Female | Soprano | 3.15 b1 | 3.87 | 3.46 | 167 | 55 |
| 3 | 25 | Female | Soprano | 3.15 b1 | 3.71 | 3.28 | 158 | 47 |
| 4 | 42 | Male | Tenor | 4.5 | 6.32 | 5.31 | 192 | 93 |
| 5 | 34 | Male | Tenor | 3.4 | 5.10 | 3.35 | 175 | 66 |
| 6 | 25 | Male | Tenor | 7.2 | 5.11 | 4.15 | 186 | 77 |
| 7 | 30 | Male | Baritone | 3.1a | 6.26 | 5.29 | 190 | 100 |

Note: Subjects 1–4 and 7 were also part of the pilot study [18].

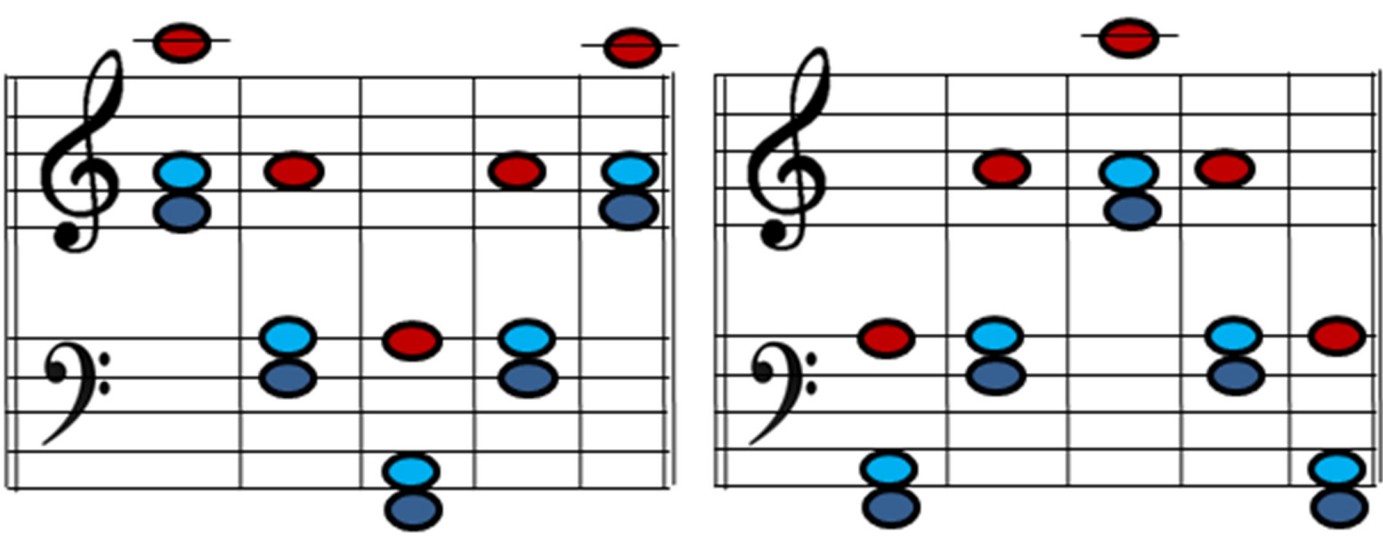

Fig 2. Musical notes and fundamental frequency ($f_o$) for task 1 and 2 for each voice classification group.

contained no irregularities such as involuntary register breaks. Each pitch was held for approximately 2–3 seconds. Thus, the phonation time was about 12–15 seconds for each task. The vowel [ɑː] was chosen throughout all measurement. Subjects were asked to sing the task in mezzo forte. The pitch was presented via headphones directly before the task sung by the investigator. The subjects could repeat the task several times until they and the investigators were satisfied.

## 2.2 Magnetic resonance imaging

The imaging of the singers'breathing apparatus was performed using a clinical 1.5 T MRI system (Tim Symphony, Siemens, Erlangen, Germany). The subject positioning and measurement was done similar to a prior pilot study [18]. For dynamic imaging, a 2D trueFISP imaging sequence (repetition time/ echo time = 3/1.5 ms, α = 6˚, bandwidth (BW) = 977 Hz/ px, slice thickness = 10 mm, acquisition matrix = 256, field of view (FOV) = 420 mm) was applied with a temporal resolution of approximately 3 frames per second (fps). Images were acquired for each task both in sagittal and coronal orientation, resulting in a total of 4 dynamic imaging series per subject (overall: 28 image sequences) which were reconstructed in real time

[32]. For the sagittal images, a slice through the right lung was chosen to avoid the stronger artefacts caused by heart motion in the left lung, which would complicate the analysis of the image. Initially, a 3D localizer data set was recorded in order to define the image plane. The sagittal plane was placed in such a way that the vertex of the DPH cupola and the apex of the lung could be identified. The coronal plane was placed similarly, encompassing both vertices of the left and right DPH cupolae and the apices of the left and right lung. All MRI measurements were recorded in the supine position during one single session. The subjects wore headphones for hearing protection.

## 2.3 Electroglottography and audio recording

The monitoring of glottal resistance during the MRI scan was performed as described before [33], using a simultaneous electroglottographic (EGG) recording with a modified MR-safe EGG device (Laryngograph Ltd. London, UK). From the EGG signal it is possible to calculate the open quotient (OQ), i.e. the ratio between the time the vocal folds are not in contact with each other and the vibratory cycle of vocal folds. Estimation of OQ was performed according to Howard et al. [34, 35] using a combination of an EGG based threshold method for detection of glottal opening (at 3/7 of the current cycle's amplitude), with detection of glottal closing instants on the dEGG (derivative of EGG) signal. Additionally, the EGG allows effective calculation of the fundamental frequency ($f_o$). Taken from a steady state portion of each pitch OQ and $f_o$ were estimated for each task from a time window of 100ms of the EGG signal. Deviations from the expected $f_o$ were calculated in cents (100 cents is one semitone) due to its logarithmic scale. The audio signal was simultaneously recorded using a microphone system (Prepolarized Free-field 1/2" Microphone, Type 4189, Brüel&Kjær, Nærum, Denmark) adapted for use in the MR environment.

## 2.4 Subglottic pressure ($p_{sub}$) and Sound Pressure Level (SPL) measurement

As the measurement of $p_{sub}$ is not adapted for simultaneous MRI and the MRI audio recording was limited due to noise interference, each subject performed the same task, as described in 2.1, directly before the MRI measurement in a sound treated room for analysis of $p_{sub}$ and SPL, also in a supine position. The instructions were the same as in the later MRI measurements but instead of a sustained vowel [a:], the repetition of syllable [pa:] was asked. The syllable [pa:] was repeated 3 times on each pitch. The audio signal was recorded at a distance of 1m from the mouth using a microphone (Laryngograph Ltd. London, UK) to estimate the sound pressure level (SPL). A calibration of the SPL was performed prior to each measurement using a sound level meter (Sound level meter 331, Tecpel, Taipe, Taiwan).

$P_{sub}$ was determined from the oral pressure during the /p/-occlusion task as described in Baken and Orlikoff [36]. Oral pressure was captured by means of a short plastic tube, with an inner diameter of 1.5 mm, mounted in a Rothenberg mask (a circumferentially vented pneuotachograph mask), so that one end was placed in the right-hand corner of the subject's mouth. Its proximal end was connected to a pressure transducer (Glottal Enterprises 162, New York, USA). $P_{sub}$ and SPL were analysed using Aeroview Version (ver. 1.4.5, Glottal enterprises, 2010, Syracuse, USA). Additionally, the EGG signal was obtained during the measurement of the $p_{sub}$ and analysed as described above.

## 2.5 MR image analysis

To characterize the motion of the lungs, distances between anatomical landmarks were manually measured in each acquired image frame (5 in sagittal and 2 in coronal images—see Fig 3

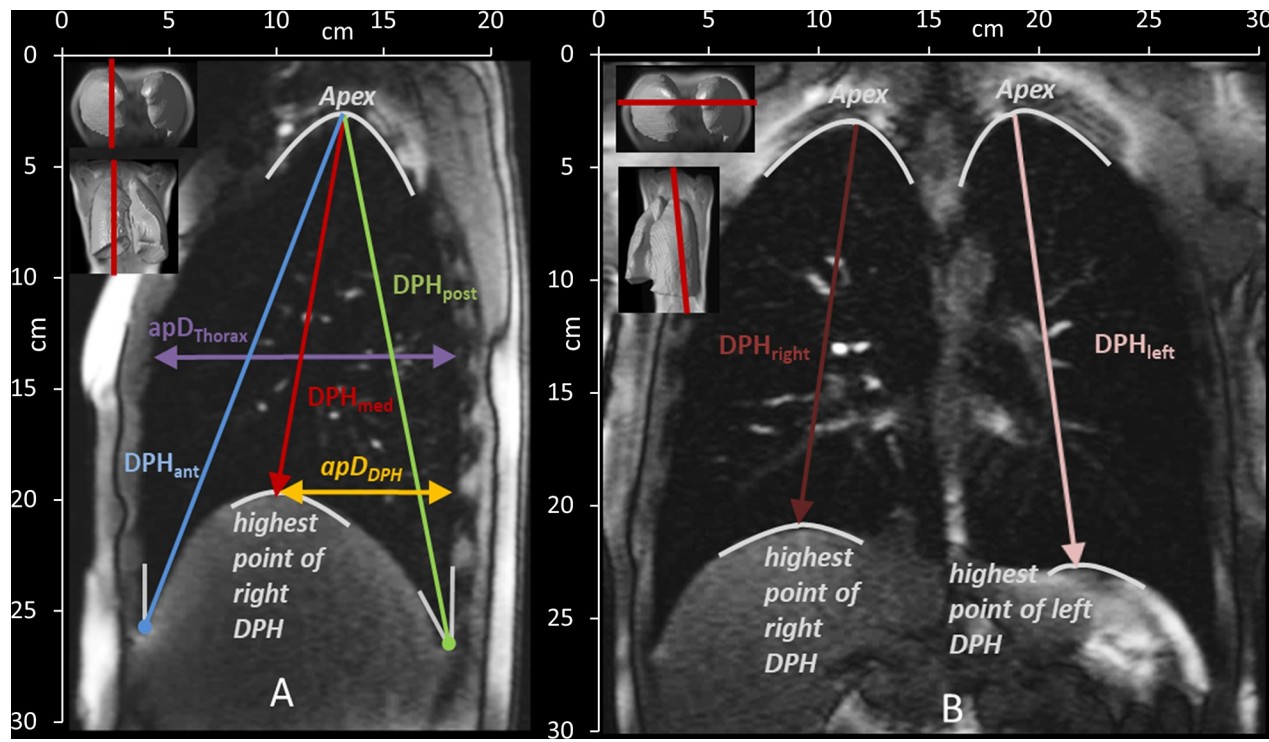

**Fig 3.** Measured distances in the sagittal (A) and coronal (B) plane and their definition according to anatomical landmarks. See also Table 2. $DPH_{ant}$ = diaphragm anterior, $DPH_{med}$ = highest point/ medial part of diaphragm, $DPH_{post}$ = diaphragm posterior, $ApD_{Thorax}$ = anterior-posterior diameter of the lung, $ApD_{DPH}$ = anterior-posterior diameter from DPH cupola to the back, $DPH_{right}$ = diaphragm right, $DPH_{left}$ = diaphragm left.

and Table 3) by one examiner (medical doctor, ENT-specialist, with many years of expertise in analysis of dynamic MRI images of the respiratory apparatus).

Thus for each subject 4 image series were analysed (coronal and sagittal for both tasks) with 7 different distances resulting in 14 different movement curves. For all 7 subjects 98 movement curves were analysed.

**2.5.1 Movement curves and normalisation.** For further evaluation of the 98 movement curves, they were plotted over time from the beginning to the end of phonation for each task. As the subjects did not sing each pitch for exact the same duration, it was not possible to directly overlay all curves for further analysis. Therefore, in a pre-processing step, the time axis was re-scaled ($t_{norm}$) according to [18], starting at the beginning of phonation ($t_{start}$) and

**Table 3. Anatomical definition of distances in the sagittal and coronal planes.**

| | |
|---|---|
| **Sagittal plane** | |
| $DPH_{ant}$ | Craniocaudal lung height from the angle of the anterior DPH and the RC to the apex of the lung |
| $DPH_{med}$ | Craniocaudal lung height from the highest point of DPH to the apex of the lung |
| $DPH_{post}$ | Craniocaudal lung height from the angle of the posterior DPH and the RC to the apex of the lung |
| $apD_{Thorax}$ | the anterior-posterior lung diameter at the height of the 5th rib |
| $apD_{DPH}$ | the anterior-posterior diameter from the highest point of the cupola of DPH to the posterior boundary of the lung |
| **Coronal plane** | |
| $DPH_{right}$ & $DPH_{left}$ | Craniocaudal lung height from highest point of DPH to the apex of the right and left lung |

ending with end of phonation ($t_{\text{end}}$). The measured distances ($A$) at different locations were normalized ($A_{\text{norm}}$) to the distance at $t_{\text{start}}$ and $t_{\text{end}}$ according to:

$$A_{norm}(t) = \frac{A(t) - A(t_{end})}{A(t_{start}) - A(t_{end})} \cdot 100$$

To compare the moments of the pitch jumps inter- and intra-individually, each jump-time-point was extracted as follows: The moment of the jump ($t_{jump}$) was defined as the 50% change of $f_{\text{o}}$ between two stable $f_{\text{o}}$s using a spectrogram of the EGG signal (obtained simultaneous during MRI) calculated with Adobe Audition (CS6, Adobe systems Inc, San José, USA). Thus, for each task, 4 jump points (e.g., high- medium, medium- low, low- medium, medium- high) were established. Around each jump-time-point, a time window of 4 frames before and after each jump ($t_{-4}$ to $t_{+4}$) was included in the evaluation (see Fig 4).

**2.5.2. Analysis of movement curves' gradient.** For statistical analysis, the gradient ($m$) of all graphs was then calculated in 8 steps ($m_{1-8}$) for the jump-time-window ($t_{-4}$ to $t_{+4}$) as the ratio of changes in measured distance over time.

$$m_n = \frac{A_{norm}(t_n) - A_{norm}(t_{n-1})}{(t_n - t_{n-1})}$$

## 2.6. Statistical analysis

The statistical evaluation is limited by the low number of participants as discussed in detail in the discussion section. The gradients of movement curves in the jump-time window were analysed in 8 timesteps ($m_{1-8}$ = factor 8) using repeated-measures ANOVAs that compare means across all variables which are based on repeated observations. Here, data of all subjects, jumps and locations was included in the calculation (see discussion section concerning limitations of the approach). To control for the bias of possible confounding variables (e.g. different subjects,

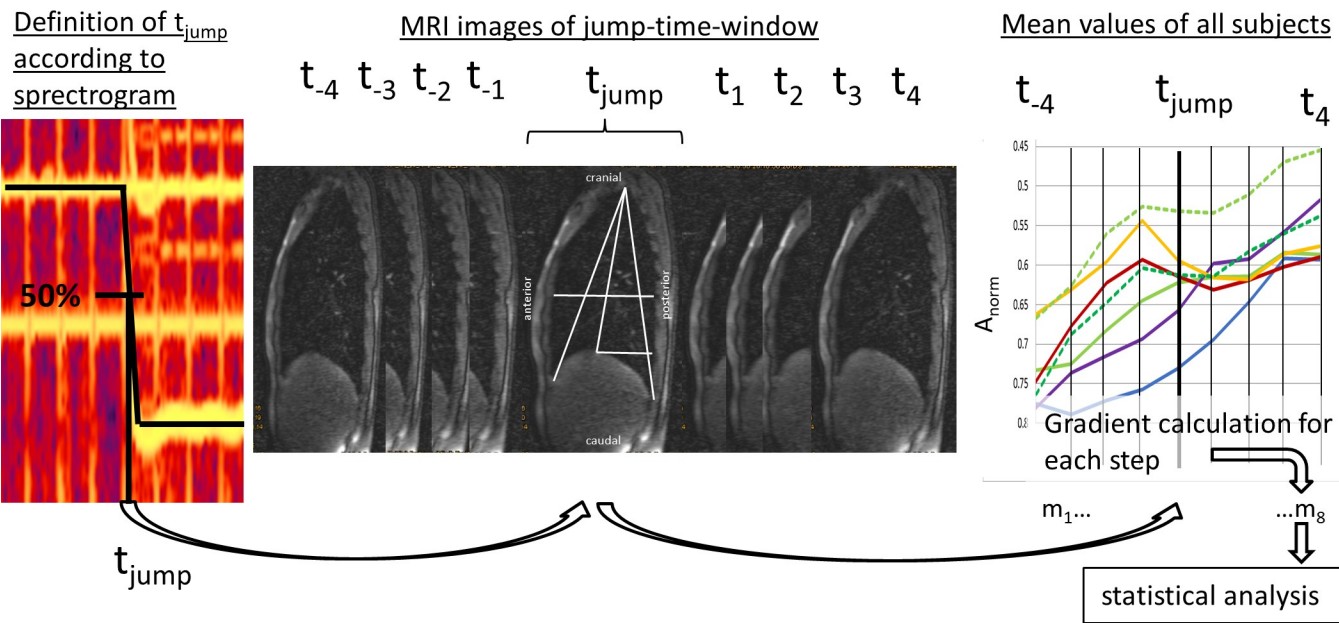

**Fig 4. Definition of jump-time-point and–window.** The jump-time-point ($t_{jump}$) was defined as 50% of $f_{\text{o}}$ differences between two different pitches. The jump-time-window was created by including 4 frames before and after $t_{jump}$ ($t_{-4}$ to $t_4$). Then mean values of distance parameters were evaluated and the mean gradient during 8 timesteps (m1-8) was derived for statistical analysis.

gender, location or tasks) they were regarded as covariates. Mauchly's test of sphericity was used to evaluate whether the sphericity assumption has been violated. As it was significant and ε >.75, the Greenhouse–Geisser adjustment was then used to correct for violations of sphericity. Shapiro-Wilk test was used to test for normal distribution of the data. It was found to be not normally distributed ($p < .001$). But, repeated measures ANOVA is believed to be very robust against normality violations [37, 38]. This calculation was first done regarding differences in jump directions (upwards vs. downwards jumps—hypothesis A). As significant differences in movement pattern occur between upwards and downwards jumps they were considered separately, for further analysis of movement curves in regard to pitch range (high vs. low jumps—hypothesis C) and jump time (early vs. late jumps—hypothesis D). To test whether different parts of the respiratory system (in terms of different anatomical locations) are possibly influenced differently during the jumps (hypothesis B) two different measures were extracted from the movement curves that optimally describe the characteristics of the movement: For downwards jumps (which present with a short inversion of the movement direction) the measure "maximal gradient" ($m_{max}$) was extracted. This value represents the maximum inversion of the movement direction. As the maximum value is not meaningful for upwards jumps which are characterized by a steady movement, here the mean gradient ($m_{mean}$) during the jump was used to evaluate whether different locations present a difference in steepness of the gradient over the whole jump window. The choice of maximum vs. mean values is clearly derived from the nature of the motion patterns. Both values were statistically tested with an univariate ANOVA. Then, statistically significant differences were further analysed with a Tukey's HSD post-hoc test. Correlations between $p_{sub}$, SPL, OQ and $f_o$ were analysed using a two-tailed Pearson correlation. For all statistical analyses, SPSS 23.0 software (SPSS, Inc., Chicago, IL) was used. The level of significance was set to $p < 0.05$.

## 3. Results

The movement patterns of the different parts of the respiratory system should always be analysed concurrently with the simultaneous status of the other key regulatory parameters. Therefore, results of the analysis of OQ, $f_o$, $p_{sub}$ and SPL during the pitch jump tasks are presented in the first part of the results section. Results of DPH and RC movement during pitch jumps from MRI images are then presented in light of the hypotheses (A-D).

### 3.1 Evaluation of OQ, $f_o$, $p_{sub}$ and SPL

OQ, $f_o$, $p_{sub}$ and SPL were analysed during sustained phonation. The mean deviation from the requested $f_o$ during the MRI was less than a quarter of a tone (*mean* = 27 cents, *SD* = 62 cents, 100 cents represents one halftone). The mean phonation time for all tasks was 10.5 s (*SD* = 1.4 s). Higher pitch correlated with a higher $p_{sub}$ ($r = .62$, $p < .001$), a higher SPL ($r = .82$, $p < .001$), a higher $p_{sub}$/SPL ratio ($r = .59$, $= p < .001$) and a higher OQ ($r = .66$, $p < .001$) (Fig 5). The amount of pressure change ($\Delta p_{sub}$) between two pitches was higher for high-medium/medium-high jumps compared to medium-low/low-medium jumps ($F(1,391) = 614.43$, $p < .001$, $^2 = .61$). In contrast, the OQ changed less for the high-medium/medium-high jumps, compared to the medium-low/low-medium jumps ($F(1,391) = 240.83$, $p < .001$, $\eta^2 = .38$) and ΔSPL did not differ significantly between the medium-low/low-medium jumps ($F(1,391) = .01$, $p = .93$, $\eta^2 < .01$). For higher pitches subjects needed a higher $p_{sub}$ for a given SPL. No significant differences were found between task 1 and 2 (early vs. late jumps) for a given pitch for OQ ($F(1, 69) = .23$, $p = .63$, $\eta^2 = .003$,), $p_{sub}$ ($F(1, 69) = 2.19$, $p = .14$, $\eta^2 = .03$) or SPL (F(1, 69) = 2.51, $p = .12$, $\eta^2 = .04$).

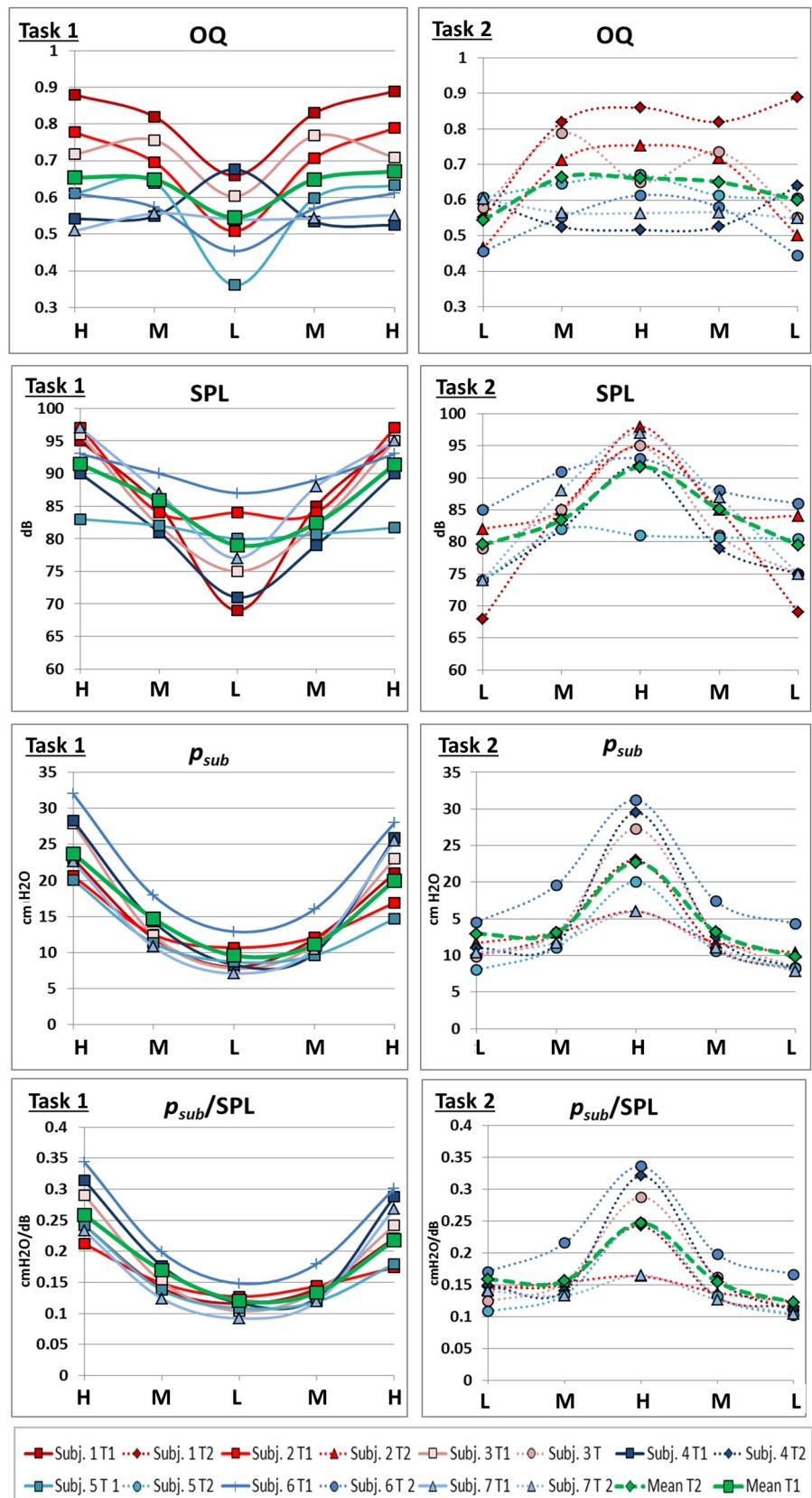

**Fig 5. Pitch correlates with Open Quotient (OQ), subglottic pressure ($p_{sub}$), Sound Pressure Level (SPL) and quotient of $p_{sub}$/SPL.** OQ, $p_{sub}$, SPL and $p_{sub}$/SPL taken from segments of sustained phonation of each pitch are displayed for each subject and task. Solid line, left side = task 1, dashed line, right side = task 2. Gender is marked with colour (red = female, blue = male). Green lines represent mean values. Please note that individual data points are presented connected for better tracing of individual points but the data points were extracted from parts of sustained phonation and not during the jump.

## 3.2. Diaphragm and rib cage movement during pitch jumps

Visual analysis of S1 Video, Fig 6 as well as all individual data in S1 and S2 Figs show a monotonic elevation of DPH (reduction of distance parameters $DPH_{ant/med/post/right/left}$) and lowering of RC (reduction of $apD_{Thorax}$) during sustained phonation with different movement velocities at different pitches. At pitch jump events, sudden inversions of the movement or the related curve gradient occur for some parts of the respiratory system. These outliers from the continuous movement always coincided with pitch jumps but do not occur for all jumps. Therefore, the movement pattern of the respiratory system during jump events was analysed further:

### 3.2.1 Curve gradient for different pitch jump directions (hypothesis A).
Measured distance data was further analysed during jump-time-windows. To test hypothesis A, movement

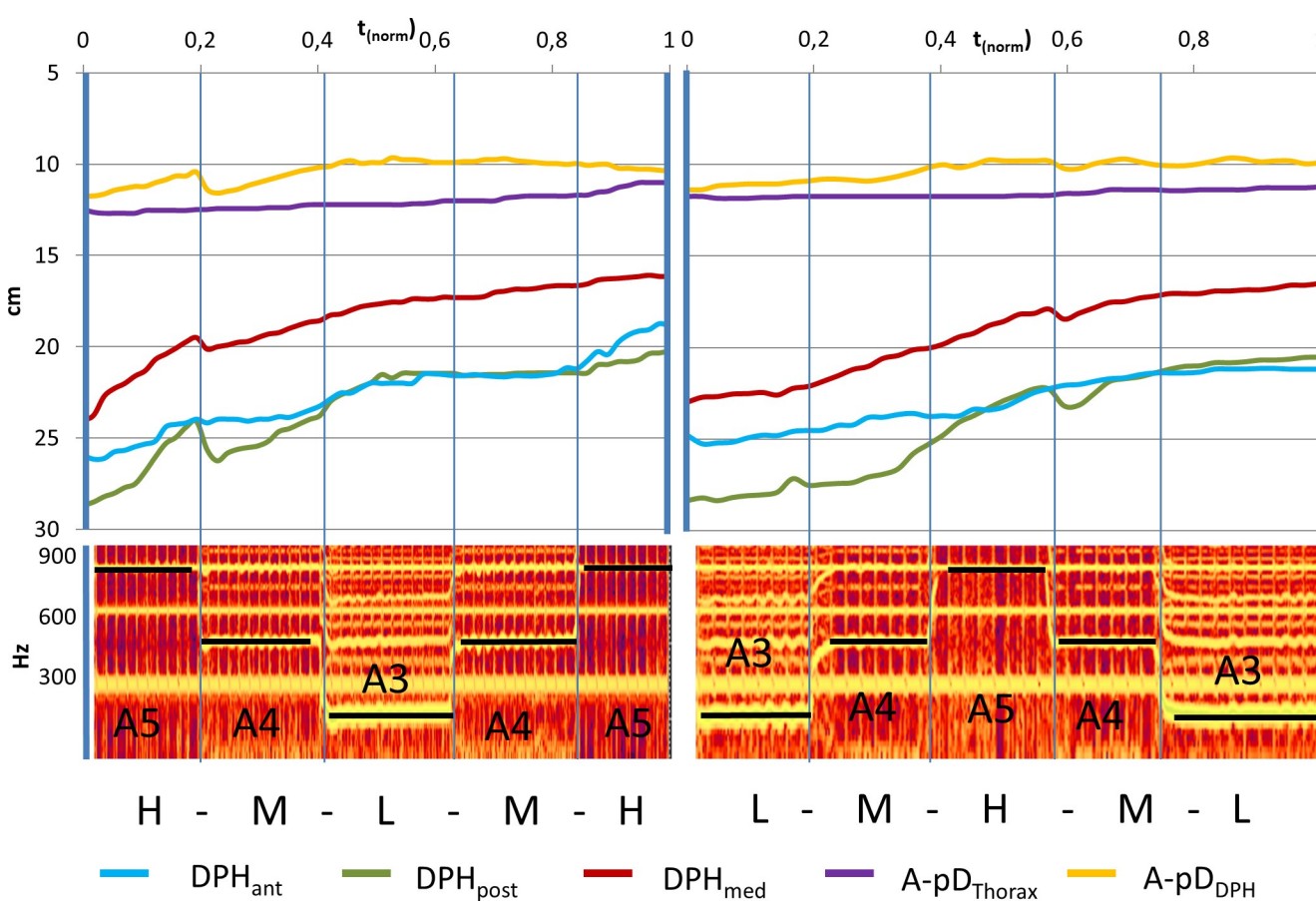

**Fig 6. Representative example curve of distance values (subject 2) with sudden inversions of curve gradient coincident with pitch jumps downwards.** Distance values are displayed in cm for all measured parameters of a sagittal image slice while singing task 1 (left) and 2 (right). Below, the corresponding spectrograph is displayed, and pitch is marked. Individual data of all subjects is displayed in S1 and S2 Figs.

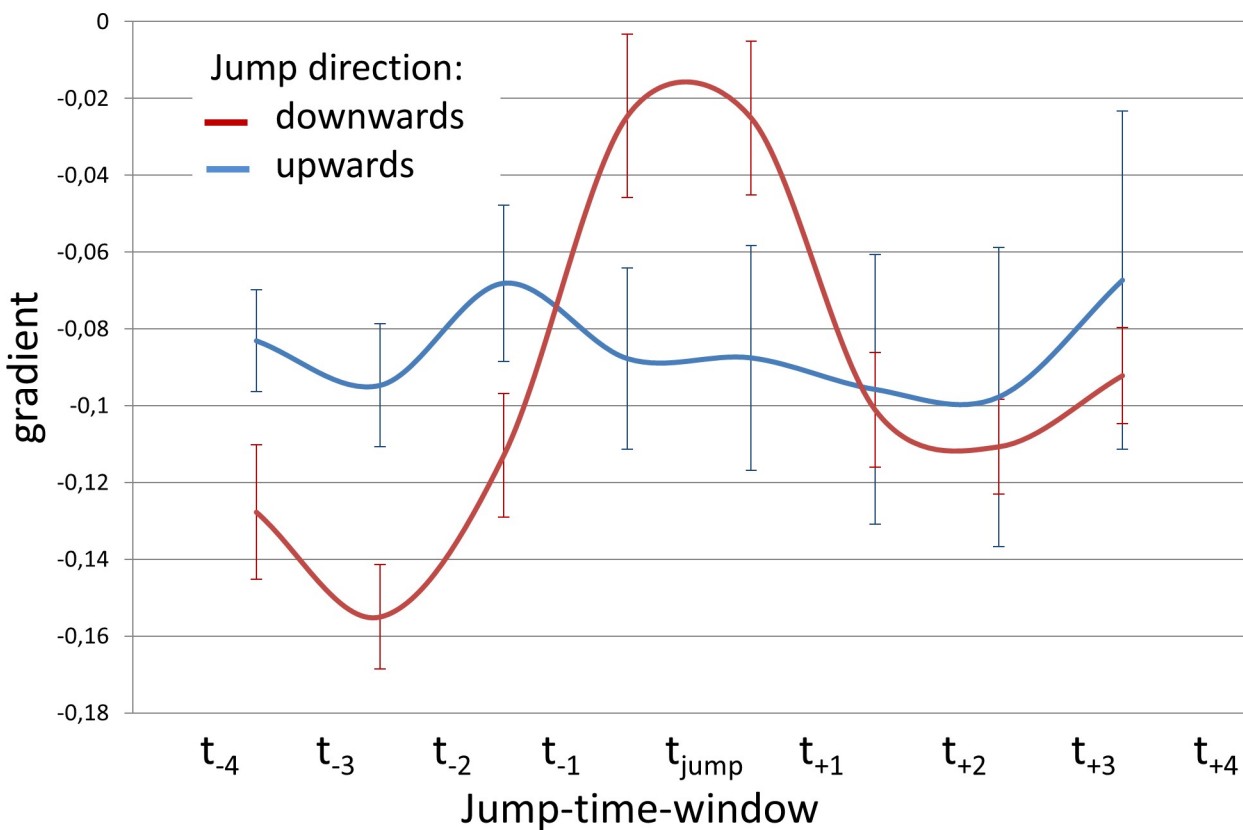

**Fig 7. Less negative mean curve gradient during downwards jumps.** Gradient of movement curves during jump-time-window are displayed for upward jumps (red) and downward jumps (blue) including standard error.

curves of all subjects and all distances during the jump-time-windows were analysed regarding the direction of the jump using a repeated measures ANOVA with the covariates subject, location and gender ($F(7, 381) = 4.39$, $p < 0.001$, $\eta^2 = .075$). This difference is characterized by a smaller negative gradient for downwards jumps at the moment of the $f_o$ change ($t_{jump}$) and can be interpreted as a temporary slowdown or inversion of the otherwise monotonic DPH and RC movement (Fig 7).

**3.2.2 Gradient curves at different anatomical locations (hypothesis B).** The described movements of the respiratory system during pitch jumps were differently pronounced at different anatomical locations as can be seen in mean curves of all subjects for different jumps (Fig 8) or the individual curves of S1 and S2 Figs. Visual analysis reveals that while some distance curves (e.g., DPH$_{ant}$) have a steady negative gradient from the beginning to the end of the jump (= monotonic/steady movement), others (e.g. apD$_{DPH}$ and DPH$_{post}$) exhibit the temporary flattening (= slowing down of movement) or even inversion of curve gradient (= movement in the opposite direction).

As shown in 3.2.1. mean movement curves differ between up- and downwards jumps. Therefore, for further evaluation of the movement characteristics of different parts of the respiratory system, up- and downwards jumps were analysed separately: As described above for downwards jumps, a sudden inversion of curve gradients occurred during the jump-time-window. Therefore, the maximum gradient ($m_{max}$) during the jump-time-window was analysed for each distance separately using an univariate ANOVA with post-hoc Tukey's-HSD (Fig 9, for all p-values see S1 Table). Analysis showed a significant difference between the different

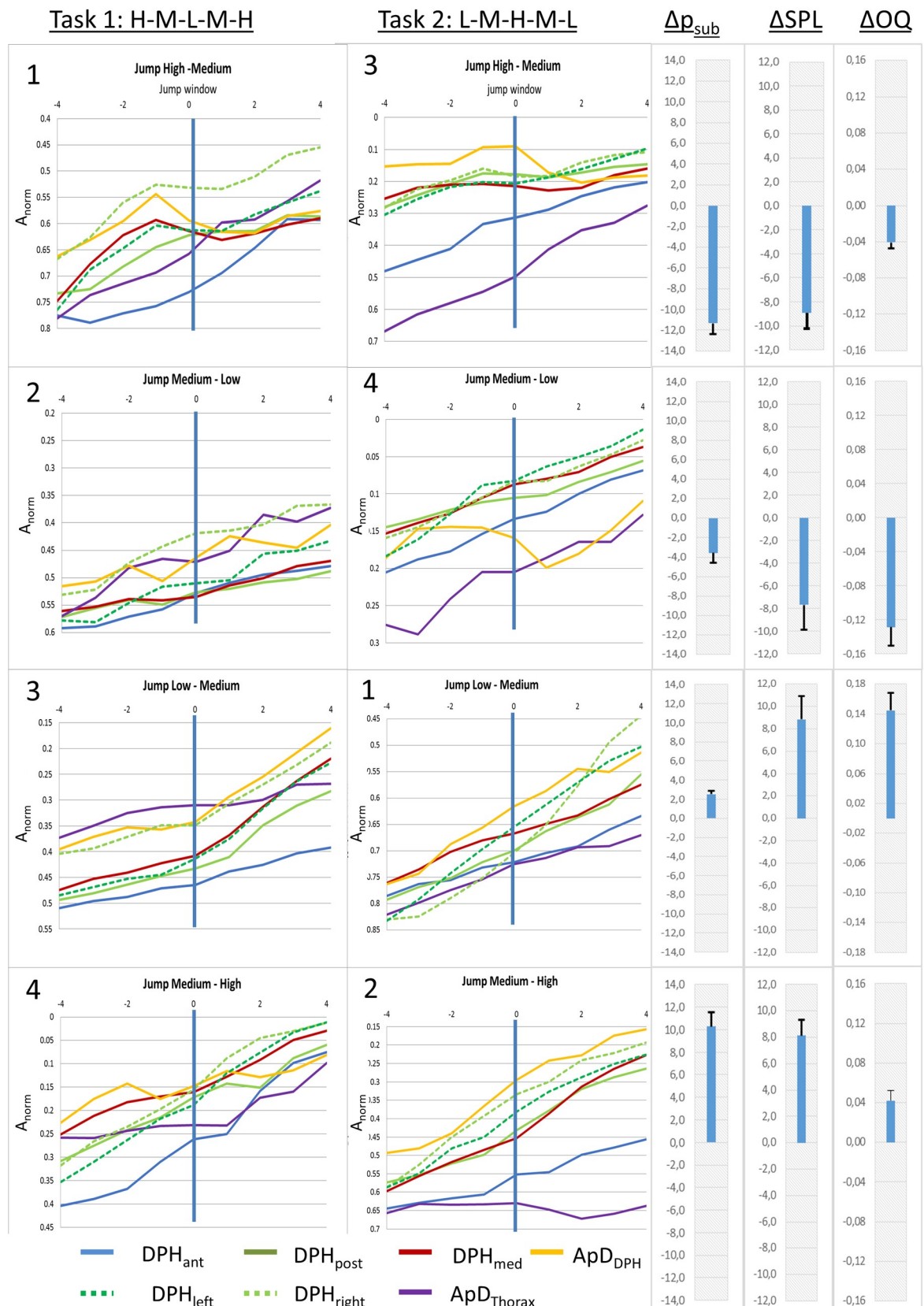

**Fig 8. Differences in movement patterns for pitch jumps between different parts of the breathing apparatus.** Mean distance curves of all subjects are displayed for jump-time-windows at all measured locations. Numbers in the upper left corner mark the order of the jumps. Additionally, mean differences in Open Quotient (ΔOQ), subglottic pressure (Δp_sub), sound pressure level (ΔSPL) between the two fundamental frequencies are displayed in the right section, including standard error.

locations ($F(6,195) = 7,19$, p < .001; $\eta^2 = .19$). The post-hoc test showed that the gradient was significantly higher for the posterior DPH (DPH_post) and anterior movement of the DPH cupola compared to all other distance parameters. The lowest values occurred for the movement of the RC (apD_Thorax) and the anterior DPH (DPH_ant). For all p-values see S1 Table.

For 6 out of the 7 subjects (except subject 7) a monotonic movement of the respiratory system was observed not only during sustained phonation but also during upward jumps. Thus, the mean gradient m_mean during the jump-time-window was analysed to assess the respiratory movements at different parts of the respiratory system (again using an univariate ANOVA with post-hoc Tukey's-HSD), which showed a statistically significant difference ($F(6,195) = 5,91$, $p < .001$, $\eta^2 = .16$). Results are displayed in Fig 10 and S1 Table. The mean gradient during the upwards jumps was significantly lower for apD_Thorax compared to all other locations except apD_DPH (for all p-values see S1 Table).

**3.2.3 Curve gradients at different pitch jump ranges (hypothesis C).** Visual analysis of Fig 8 revealed that the amount of short-term gradient change for high-medium jumps was greater compared to medium-low jumps. To test hypothesis C a repeated measures ANOVA with the covariates subject, location and gender was used with data for pitch jump range treated separately for upwards and downwards jumps. Analysis of the gradient curve progression in the jump-time-windows revealed a statistically significant difference for high-medium vs. medium-low for jumps in a downwards direction ($F(7,185) = 2.90$, $p = .007$, $\eta^2 = .099$; see Fig 11A). This difference did not reach statistical significance in upwards jumps ($F(7, 185) = .82$, $p = .57$, $\eta^2 = .03$).

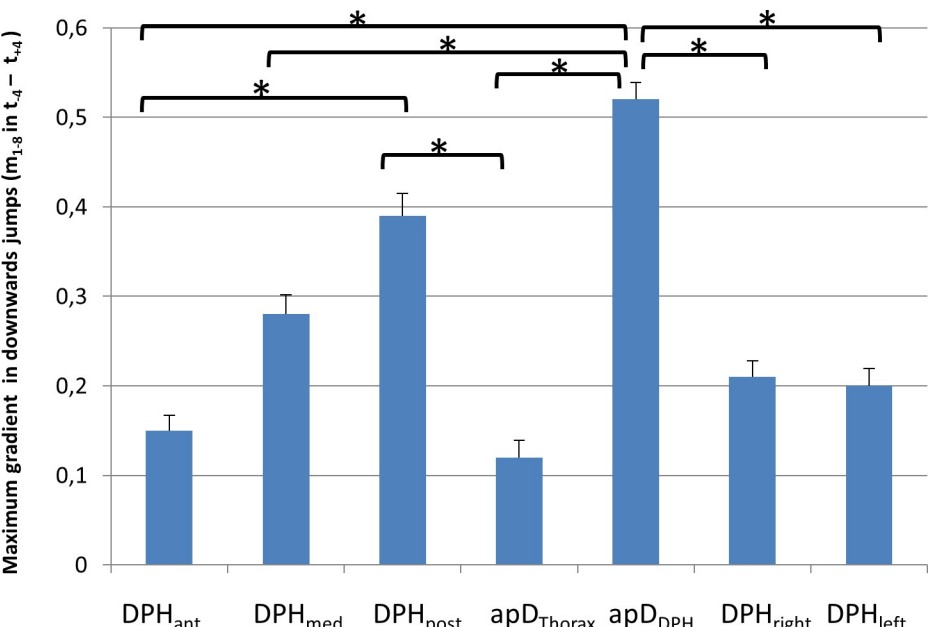

**Fig 9. Maximum gradient (m_max) is higher for posterior diaphragm (DPH_post) and anterior movement of the diaphragm cupola (apD_DPH) compared to all other parts of the respiratory system.** m_max is displayed for downwards jumps with standard errors. Significant differences are marked (* < .05), for all p-values see S1 Table.

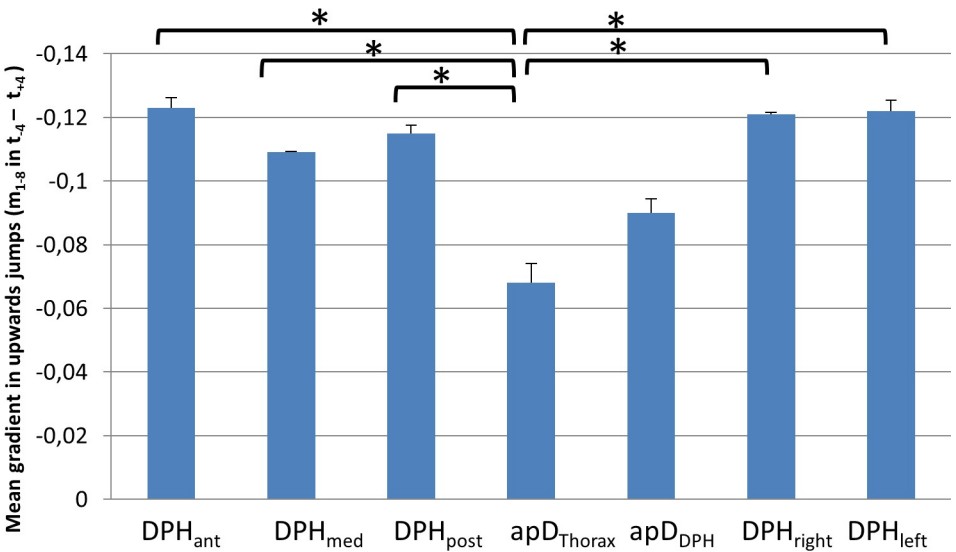

**Fig 10. Lower mean gradient during upwards jumps for thorax diameter (apD$_{Thorax}$).** Mean gradient during the jump window for upwards jumps at different locations with standard deviation is displayed. Significant differences are marked (* < .05), for all p-values see S1 Table.

**3.2.4 Gradient curves for different pitch jump-time points (task 1 vs. task 2, hypothesis D).** Tasks 1 and 2 comprised the same jumps, but in a different order. Thus, the same jump occurred in one task earlier (thus on higher lung volume) and in the other task later. Jumps could therefore be separated into early and late jumps. To test hypothesis D a repeated measures ANOVA with the covariates subject, location and gender according to the time of the

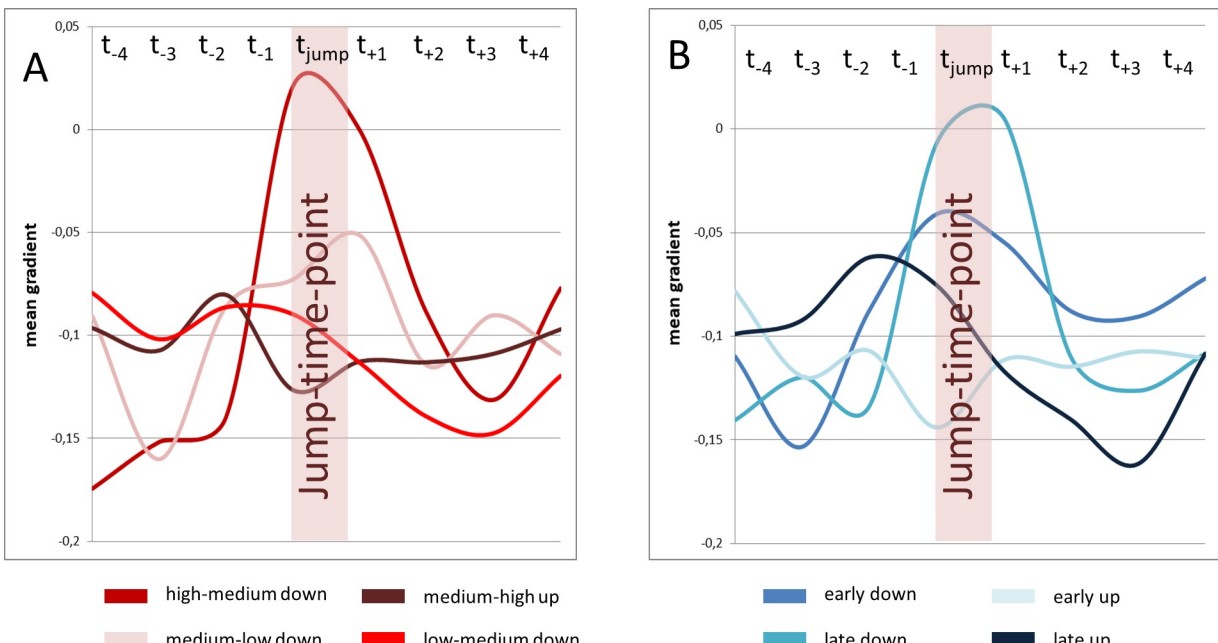

**Fig 11. Gradient change at high-medium jumps was pronounces compared to medium-low jumps, while no difference occurred between early and late jumps.** Mean gradient curves (all subjects and all distance parameters) for different jump events are displayed. High-medium and medium-low jumps are displayed left (A), early and late jumps are displayed right (B).

jump was used. For downwards jumps, early and late jumps did not differ in the visual analysis of curve progression in Fig 11B. Additionally no statistically significant difference between the gradients of early and late downward jumps was found ($F(7, 185) = 1.34$, p = 0.24, $\eta^2 = .05$). For upwards jumps a pre-jump gradient change for late upwards jumps occurred that could not be seen in early upwards jumps (see Fig 11B). Statistical analysis of the gradient curve also revealed a statistically significant difference in curve progression in this case ($F(7, 185) = 2.97$, $p = .006$, $\eta^2 = .10$).

## 4. Discussion

In the presented study, movements of the breathing apparatus for $p_{sub}$ control during pitch jumps were analysed in 7 professionally trained singers. Measured distances in the respiratory apparatus reduced monotonically over time during phonation, presenting a general movement pattern that was similar to the phonation of sustained pitches [18]. The monotonic movements were interrupted by sudden inversions of movement direction during the jumps. Differences relating to jump direction, $f_o$ range and anatomical location became apparent through further analysis.

Throughout sustained phonation, the singers´ DPH was elevated and the RC diameter was reduced in a monotonic manner. During upward pitch jumps the DPH was raised more quickly compared to sustained phonation, which can be regarded as contributing to the pressure generation for the new higher pitch. In contrast, for downward pitch jumps the DPH frequently moved downwards (in an inspiratory direction) or its elevation slowed down during the jump. As $p_{sub}$ is correlated with $f_o$ [39], pitch jumps are frequently associated with a sudden adaptation of $p_{sub}$ for the new pitch. This was also confirmed in the presented data. In the current study $p_{sub}$ could not be measured during the MRI scan (due to noisy surroundings) but was acquired in a separate session on the same day. As professional singers have very consistent breathing patterns [28], it can reasonably be assumed that the $p_{sub}$ during measurement is consistent with that during the MRI scan. Still it must be mentioned that the task for acquiring $p_{sub}$ (syllable repetition of [pa:]) is different from the MRI task (sustaining of vowel [a:]). Whether the p/-occlusion would affect $p_{sub}$ cannot be answered with the last certainty.

It is reasonable to assume that the short-term deceleration or inversion of the direction of the DPH movement is associated with the required reduction of $p_{sub}$ for the downward jumps. The sudden activation of inspiratory muscles during phonation for downward pitch jumps was postulated by Leanderson et al. in a study with 4 professional male singers [5]. Their study analysed the transdiaphragmatic pressure as a summative measure for DPH activity during phonation of pitch jumps and showed a sudden DPH contraction for downward pitch jumps in 3 out of 4 professional singers. This is in accordance with the presented data and as stipulated in hypothesis A, the movement curves of downwards jumps differed from upwards jumps in most subjects (DPH contraction vs. constant elevation). But, in contrast to transdiaphragmatic pressure measures which only analyses a cumulative effect of DPH and RC movement, differences according to anatomical locations of the respiratory system could also be analysed in the current study from the MRI based data.

The sudden inspiratory movements were not evenly distributed for all parts of the respiratory system but were focused on the most posterior DPH and associated with an anterior movement of the DPH cupolas, which are typical signs of a DPH activation [25]. This is underlined by differences in $m_{max}$: this measure indicates the most vigorous movement in a positive (inspiratory) direction during the jump windows and revealed a significantly higher value for the anterior movement of the DPH cupola ($apD_{DPH}$) and the downwards movement of the posterior DPH ($DPH_{post}$) compared to all other parameters (Fig 9). However, the DPH

activation is not transmitted to its most anterior part, which, alongside the RC was also continuously reduced during jumps without contraction. This supports data derived from ultrasound measures in 3 professional singers during phonation of scales [15]: The authors describe differences in the movement pattern of the dorsal DPH section, with a bumpy movement pattern at ascending pitch, compared to a close to linear pattern of movement of the anterior sections.

Thus, in the current study, in the breathing apparatus of the professional singers two independently controlled functional units could be demonstrated during pitch jumps downwards. The same separation of functional units was observed for sustained phonation [18]: here, in the first phase of phonation, movements of the respiratory system occurred mainly in the back part of the DPH while the RC and anterior DPH were stabilized in a more inspiratory position changing to the opposite for the last part of phonation with a quicker movement in the RC and anterior DPH. The close attachment of the anterior DPH to the RC and the smaller movement range of the anterior compared to the posterior DPH [18] might be the origin of the different movement patterns between the anterior and posterior DPH. Compared to the DPH the RC wall has a higher impact on pulmonary air movement due to its larger contact area [25]. Therefore, at a given glottal resistance the RC has a greater effect on $p_{sub}$ for the same amount of RC and DPH movement. For singers, a very constant movement of air and close control of $p_{sub}$ is mandatory to stay in tune and to sing with the intended loudness. Therefore, it might be more efficient to keep the RC with the anterior attached DPH more constant and perform quick $p_{sub}$ adaptations with the posterior part of DPH. Here, the movement range is 4–5 times greater compared to the RC, and 2 times greater compared to the anterior DPH [18] which might allow for a more precise $p_{sub}$ adaptation.

For upward jumps, a higher $p_{sub}$ is necessary for the higher $f_o$. In the data this was primarily associated with an elevation of the DPH, and to a lesser degree via RC contraction. The DPH elevation is usually initiated by compression of the abdominal compartment by the abdominal wall muscles (AW) and a relaxation of the DPH [25]. However, especially in one male subject (see S2 Fig, subject 7, task 1, jump F3-F4) the DPH also performed a short contraction movement during the upward jump. Similar movement patterns were also described in transdiaphragmatic pressure measurements in a single male subject by Leanderson et al. [5]: This professional singer forcefully contracted his AW muscles during phonation and reduced the pressure constantly by increasing his tonic DPH activity. For short term adjustments of $p_{sub}$ in pitch jumps the singer shortly activated the DPH for both upward and downward jumps. It was speculated that this was done to counteract the forceful AW contraction for fine control of $p_{sub}$ adaptation. This was described as DPH-co-contraction technique, in contrast to the flaccid DPH technique, where the DPH was only activated during phonation for reduction of $p_{sub}$. Thus, the different behaviour during upward jumps in our subjects could also indicate different strategies in accordance with Leanderson et al. [5].

$P_{sub}$ analysis showed, in agreement with existing literature [26], a significantly greater difference from high-to-medium $f_o$ jumps compared to medium-to-low $f_o$ jumps. Supporting hypothesis C, the DPH contraction was also significantly more vigorous (Fig 11A) for high-medium jumps compared to medium-low jumps. It can be assumed, that the described movement of the DPH to reduce $p_{sub}$ in downwards pitch jumps is economic in western classical singing as it was documented in professional trained singers in this study. Whether it also occurs intuitively in untrained subjects or if the lack of this movement is associated with voice disorders is not investigated so far. It could be speculated that a failure to reduce the $p_{sub}$ in pitch jumps downwards by the breathing apparatus could be associated with singing out of tune or louder then intended. However, also glottal, or vocal tract adaptions affect and regulate $p_{sub}$ [2]. Malregulation of the movements of the respiratory system could therefore be related to the necessity of adaptations on glottal or vocal tract level or the other way around. The

chosen tasks and pitch represent the whole tessitura of the singers. It can therefore be assumed that different registers functions were used by the singers in the way they would do it on stage [40–42]. The vowel [a:] was chosen in all tasks to avoid the articulatory effects that can be expected when fundamental frequency exceeds the normal value of the first format [43, 44]. Still, also articulatory adaptations like formant tuning for high phonation in soprano voices could influence the vibration of the vocal folds and thus $p_{sub}$ [44–47]. Whether the DPH contraction is also related to specific register functions was not investigated in the presented data but could be of interest in further evaluations in that theme.

According to literature on breath support [4, 39, 48], a maximally inflated lung and thorax leads to a passive exhalation force (recoil force) of approximately 30 cmH$_2$O when the vocal folds are adducted for phonation. When this pressure is too high for the intended phonation, it has to be reduced by a contraction of the inspiratory muscles at the beginning of phonation. The need for this activity then gradually decreases with lung volume up to the point where the passive exhalation forces cease (resting expiratory level, REL). Beyond REL the expiratory muscles have to compensate for the growing inhalation force of the increasingly compressed RC and lungs. Thus, the passive pressure situation due to recoil forces is changing fundamentally from the beginning to the end of phonation. It was hypothesised (3) that the time of the jump (the same jump occurred at the beginning in task 1 and end in task 2 –and vice-versa) would lead to differences in the motion curve. In contrast to the hypothesis no significant difference was found in the gradient analysis for the same downwards jump in task 1 and 2. Similarly, no difference was found for OQ, $p_{sub}$, SPL for the same pitch between task 1 and 2. This stands in contrast to Leanderson et al. who found that for octave jumps over a maximum phonation time, the DPH contractions occurred more vigorously at the beginning compared to the end [5]. Additionally, Iwarsson et al. found, an increase in closed quotient with decreasing lung volume, while subglottal pressure decreased [49]. As the task of the presented study was clearly shorter than a maximum phonation time, the difference might not be so pronounced in our data.

A major limitation of this study is that the measurements were taken in supine body position due to the use of a clinical horizontal-bore MRI system. Studies on posture-related differences showed that in normal breathing the DPH-motion in the supine position was significantly greater than that in the upright position [20]. Also, functional differences have been reported between upright and supine breathing (in supine position functional residual capacity increases [50, 51], vital capacity and forced VC [52] decreases). Furthermore, a more forceful contraction of the DPH during inhalation and a less forceful contraction of the abdominal wall during phonation was observed in the supine position [53]. This could be due to the fact that, while lying on the back, gravity adds force on the lungs and thus to $p_{sub}$, so that the demand for raising $p_{sub}$ by muscular means is smaller. It can be stated that gravity acts as an inspiratory force on DPH and AW and as an expiratory force on RC in an upright position. In the supine position, however, the gravitational force would change to a more expiratory direction. This is in accordance with the data of Hixon et al. for speech phonation [54], who described a greater passive pressure contribution of the RC in supine phonation. For speech phonation he also described major differences for the inspiratory effort of the chest wall, which was provided mainly by the RC for upright and mainly by the DPH for supine phonation [6]. Nevertheless, studies on VT configuration in professional singer vs. untrained subjects have found that the posture effect was systematic and small for professionals [55] and greater and random for untrained subjects [56]. The great expertise and highly controlled breathing apparatus of a trained singer might also be less influenced by posture compared to untrained subjects, as professional singers today are used to singing in different body positions. The exact influence of body position on DPH motion cannot yet be understood by

means of imaging. Findings of a small pilot study with a rotatable MRI scanner rather support a systematic difference without a fundamental change in respiratory dynamics [57] but further studies are necessary. As explained above, during singing $p_{sub}$ has to be continuously adapted for singing to be in tune—the singers succeeded in this regard as the mean frequency difference was less than a quartertone. Further MRI studies with a rotatable MRI device (e.g. comparable to [55]) would be helpful to clarify the posture dependence.

To minimize potential distractions in the noisy MR environment, professionally trained singers were included in the study–however, the influence of the Lombard effect [58] on phonation cannot be totally excluded. The phonation time was about 15 seconds per task with each pitch held for about 3 seconds. The pitch jumps were sung in a "legato" way and were performed very rapidly. The time between the two frequencies lasted less than a second. Therefore, the temporal resolution of 3 frames per second could be limited for very fast changes in the respiratory apparatus.

Additionally, the following limitations of the statistical evaluation of the presented data must be mentioned: The approach to statistical analysis is clearly limited by the few participants. However, as professional singers are a very special group of participants the number could not be raised. For statistical evaluation, the data of curve gradients ($m_{1-8}$) was pooled and analyzed using repeated measures ANOVA. This approach violates the independence of observations. This is a clear limitation, but it is accepted also in other fields, when data from re-tests are included as independent values, when the number of participants cannot be raised. Additionally, the presented data of curve gradients ($m_{1-8}$) is not normally distributed. This is probably also originated in the paucity of subjects. But fortunately repeated measures ANOVAs are believed to be very robust against against normality violations [37, 38]. The presented study analyzed professional singers' behavior of the respiratory system during pitch jumps by direct visualization and enabled a detailed observation of the diaphragm contraction during pitch jumps downwards for the first time. But the study design also revealed that this phenomenon is related to different requirements like the amount of subglottic pressure difference. That complicates the evaluation by only visual analyzation. Thus, in the author's opinion, the statistical evaluation helps the reader to follow the evaluation related to the presented hypotheses. Even if the results of the statistical approach are only of minor indicative value, they are helpful to get a more distinct idea of which effect should be closer analysed in a bigger cohort in the future.

## 5. Conclusion

In contrast to sustained phonation, where DPH and RC move only in an expiratory direction, singers regularly activated inspiratory breathing muscles for phonation of downward pitch jumps with a sudden reduction of $p_{sub}$ while phonation continued. This is to the best of the authors' knowledge the first study to visualize these regulatory movements in professional singers using dynamic MRI. The advantage of MRI is that the movement of different parts of the respiratory system can be analysed independently. This revealed that the inspiratory movement during pitch jumps downwards was primarily executed in the posterior part of DPH. It was associated with a ventral movement of the DPH cupolas with simultaneous continued expiratory movement of the RC and anterior DPH. Thus, the RC/ anterior DPH and medium/ posterior DPH can be regarded as different functional units during respiratory regulation of $p_{sub}$, which is in accordance with the regulation of sustained phonation [18]. It seems favourable for singers to use the more flexible posterior part of DPH for the fine control of $p_{sub}$ whilst maintaining a constant movement in the RC/ anterior DPH. The magnitude of the adaptation movement was related to the phonated $f_o$ and thus the level of the $p_{sub}$ differences but was not dependent on the time point of the jump.

DPH displacement in a cranial direction primarily provided the force generation to increase $p_{sub}$ for upwards jumps but different strategies became apparent between subjects. A larger cohort is necessary to better understand possible differences in breathing strategies related to sex, fach (classification of singers according to range, weight, and colour of the voice) or musical genre. Also, the impact of gravity on the respiratory system during singing needs to be studied. Improving our understanding of the phonatory breathing processes of the respiratory system utilised by highly trained voice users could help to identify and improve less efficient strategies e.g. in patients with voice disorders in the future.

## Supporting information

**S1 Fig. Individual movement curves of all female subjects and distance parameters with normalized phonation time.** Different $f_o$s are marked with different shade taps indicating the jumps (darker shape higher $f_o$ and lighter lower $f_o$).
(TIF)

**S2 Fig. Individual movement curves of all male subjects and distance parameters with normalized phonation time.** Different $f_o$s are marked with different shade taps indicating the jumps (darker shape higher $f_o$ and lighter lower $f_o$).
(TIF)

**S1 Table. p-values for differences in $m_{mean}$ at different location for jumps upwards (red boxes) and $m_{max}$ for jumps downwards (blue boxes).** Significant differences ($p < .05$) are marked with darker colors.
(DOCX)

**S2 Table. Data table including subject number, gender, jump, location, gradient in m1-m8, delta $p_{sub}$ *(subglottic pressure)*, delta OQ (open quotient), delta SPL (sound pressure level).**
(XLSX)

**S1 Video. Example video (subject 2) of dynamic Magnetic Resonance Imaging (MRI).** The subject sings task 1 & 2. Sagittal image slices are shown as a high contrast version of the MRI based images in the right part of the screen. In the left part of the screen distance measures and spectral analysis of the filtered MR audio signal are presented. The red line indicates the progression of the task simultaneous to the MRI film.
(MP4)

## Acknowledgments

The authors thank Dr. Helena Daffern, PhD, for native correction. The authors thank Dr. Manfred Nusseck, PhD and Dr. Nico Hutter, PhD, for their help with statistical evaluation. The authors would also like to thank the subjects for their willingness to take part in this study. Part of the material was presented at the UEP Conference 2018 in Helsinki.

## Author Contributions

**Conceptualization:** Louisa Traser, Fabian Burk, Michael Burdumy, Bernhard Richter, Matthias Echternach.

**Data curation:** Louisa Traser, Ali Caglar Özen, Michael Bock, Daniela Blaser.

**Formal analysis:** Louisa Traser, Daniela Blaser, Matthias Echternach.

**Funding acquisition:** Bernhard Richter.

**Investigation:** Louisa Traser, Fabian Burk, Michael Burdumy, Michael Bock, Matthias Echternach.

**Methodology:** Louisa Traser, Ali Caglar Özen, Michael Burdumy, Daniela Blaser.

**Project administration:** Louisa Traser.

**Software:** Ali Caglar Özen, Michael Burdumy, Michael Bock.

**Supervision:** Bernhard Richter, Matthias Echternach.

**Writing – original draft:** Louisa Traser, Matthias Echternach.

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
