## [Decision Letter · Decision Letter 0]

2 Dec 2019

PONE-D-19-28236

Respiratory regulation of subglottic pressure for phonation of pitch jumps – a real-time MRI study

PLOS ONE

Dear Dr. Traser,

Thank you for submitting your manuscript to PLOS ONE. After careful consideration, we feel that it has merit but does not fully meet PLOS ONE’s publication criteria as it currently stands. Therefore, we invite you to submit a revised version of the manuscript that addresses the points raised during the review process.

Please carefully answer to the issues raised and especially to the major issues raised by revier 1 and 3 (e.g. statistics, readability of manuscript, data ised in previous study, ...)

We would appreciate receiving your revised manuscript by Jan 16 2020 11:59PM. To enhance the reproducibility of your results, we recommend that if applicable you deposit your laboratory protocols in protocols.io, where a protocol can be assigned its own identifier (DOI) such that it can be cited independently in the future. For instructions see: http://journals.plos.org/plosone/s/submission-guidelines#loc-laboratory-protocols

We look forward to receiving your revised manuscript.

Kind regards,

Michael Döllinger, Ph.D.

Academic Editor

PLOS ONE

Journal Requirements:

Additional Editor Comments (if provided):

After reading the work myself, I recommend to please especially respond to the issues raised on statistics, readability and previous study.

Reviewers' comments:

Reviewer's Responses to Questions

**Comments to the Author**

1. Is the manuscript technically sound, and do the data support the conclusions?

Reviewer #1: Yes

Reviewer #2: Yes

Reviewer #3: Partly

2. Has the statistical analysis been performed appropriately and rigorously? 

Reviewer #1: No

Reviewer #2: Yes

Reviewer #3: No

3. Have the authors made all data underlying the findings in their manuscript fully available?

Reviewer #1: No

Reviewer #2: Yes

Reviewer #3: Yes

4. Is the manuscript presented in an intelligible fashion and written in standard English?

Reviewer #1: No

Reviewer #2: Yes

Reviewer #3: No

5. Review Comments to the Author

Reviewer #1: The study is an analysis of respiratory function during the phonation of pitch jumps using real time magnetic resonance imaging. Professional singers sang sequences of notes from low through high back to low in one imaging run and the reverse pattern in another run. Interest was primary in the transition between pitches. Both were repeated for different angles of image acquisition that were useful for measuring different anatomical structures. Images were used to measure distances between anatomical landmarks within the lungs as a proxy for lung volume and to understand the separate contributions of various respiratory muscles to controlling subglottal pressure for sound production. This study has the potential to provide information on a muscular system that has been understudied. However, while I find the experimental methodology to be sound, I have serious concerns about the statistical analysis.

Major concerns

1) The authors refer to a pilot experiment that has previously been published using a similar technique on an overlapping sample of participants. It is important that the introduction clearly inform the readers on the difference between these two studies and the degree to which the present study is a novel extension versus a replication of previous findings. Though replications are very valuable I am concerned about the fact that the same participants were used in both experiments and that the authors may have sliced their data too thinly.

2) From visual inspection of the data the authors have decided to analyze the maximum change in distance between landmarks for some conditions (figure 8) or the mean change is distance between landmarks in others (figure 9). Since this choice was made post-hoc the authors have implicitly performed all pairwise tests within both measures, and should correct for them. Whereas present they appear not to perform any corrections for multiple tests. All analyses for both of these figures should be corrected for the ~40 separate tests that the authors have conducted (more if they have also visually inspected other possible measures such as the minimum distance). Stats for all of these tests should also be reported explicitly in a table.

3) Throughout the results section the authors describe the findings in terms of contrasts (in such a condition X measure Y was lower than in condition Z). However, they report statistical tests that appear to be omnibus anovas, which do not permit this interpretation. Please provide more careful interpretations of the statistics that are report or choose statistical tests that are more closely aligned with the inferences that interest the authors.

4) The authors should also report how they have handled non-independence in their data. Each task contained multiple pitch jumps of the same type, and it would be inappropriate to treat these replicate jumps as independent when they come from the same singer. It would be appropriate to use statistical methodology that can cope with this, such as linear mixed models, or to simply average across replicate jumps. From the very large degrees of freedom reported for the omnibus anovas, I suspect that this was not done. Even so I am confused by how large the degrees of freedom is here and would appreciate some clarification of what went into this analysis.

5) Please provide the underlying data as per plos1 policy. Currently, only a subset of the data are provided and only for one participant. It would be most appropriate to include the raw MRI images organized in some sensible directory tree as well as all derived data.

Minor concerns

abs) There should be no line breaks in the abstract

abs) References to theory should come early in the abstract not late

intr) The biomechanical review at the beginning of the introduction would benefit from an explanatory figure for non-specialists

l53) the previous ultrasound study should be described in more detail, as with the pilot study

l93) "permanent" may not be the correct word

l118) please report this phoneme in IPA format

l153) extra period

l163) what where the instructions for the out of scanner recordings? Should we be concerned about gross differences in SPL or Psub if singers were directing themselves towards a person or microphone that was farther away in the out of scanner test versus a microphone that is nearby in the scanner as well as the sense of being alone that one has in that context?

l199) remove the % sign from the equation

l217) Something doesn't seem right in the denominator. Should you really be dividing by a larger number as you step through frames?

l226) if the threshold was p = 0.05 then do not report p < 0.01 in figures. P-values are never a measure of effect magnitude. The same applies to subsequent figures.

Fig4) remove interpolation between points as you probably don't mean to imply that there are smooth transitions as depicted

l252) correct several typos in this section

Figs 6 & 7) The means in these figures are not interpretable without error bars.

Figure 9)

l313) why have the authors switched from reporting effect sizes in eta squared to cohen's d? Is this intentional or a typographical error? Also, these effect sizes appear larger than I would expect for non-significant effects with the degrees of freedom report here. Please double check these values.

l243) adaption is not an english word, please revise here and throughout the discussion.

l343) remove extra space

l408) remove the (= note) notation here and throughout

l449) please clarify what you mean by rotatable MRI device

l472) please remove the // notation and provide full sentences

Reviewer #2: This is a very interesting and well-conducted study. I have only minor comments.

1. Line 32-33, and Line 337. "elevated" and "reduced" need to be referenced to some baseline. Elevated and reduced compared to what?

2. Line 37. "rip cage" should be "rib cage".

3. Line 38. "Diaphragm elevation primarily generated the force to increase subglottic pressure", but the diaphragm is an inspiratory muscle, and its elevation does not generate the force, rather, it is probably abdominal muscles that generate the force and elevate the diaphragm.

4. Line 40. "underlines" should be "underline".

5. Lines 84-96. I think the hypotheses could be highlighted more clearly, perhaps using a list format rather than a paragraph format. They should also be brought back to mind in the Results section.

6. Line 102. I'm not sure it is a valid assumption that professional singers would be "less distracted by the noise".

7. Lines 106-107. There is no explanation of how the VC and FEV1 measures were obtained. The VC measures in Table 1 are quite large - is it accurate?

8. Line 113. The pilot study has a very similar title. How is it different from the present study, beyond the addition of 2 subjects?

9. Line 119. "with no pitch repetitions". What does this mean?

10. Line 123 and Lines 127-135. Is the method employed here rtMRI or is it truly real time MRI, since the notes are held for so long?

11. Line 139 and Line 140. "coupola" should be "cupola".

12. Line 150. "detection of glottal opening (at 3/7)". What is "(at 3/7)" mean?

13. Lines 152-153. "estimated for each task ... from a steady state portion of each pitch." I found this sentence a bit awkward. There is an additional period "." at the end.

14. Line 170. "Rothenberg mask". It's not clear to me how this was used.

15. Line 202. "EGG-MRI signal". What does this mean?

16. Lines 235-246. It's not clear where the degrees of freedom of "391" and "69" are coming from in the statistical analyses. Also, Line 241 has "614,43" which should be "614.43".

17. Line 249. "right side = task." Should be "right side = task 2."

18. Line 253. This sentence is a bit awkward.

19. Line 255. "distances.. These outliners" should be "distances. These outliers".

20. Line 261. "correlating spectrograph" should be "corresponding spectrogram".

21. Figure 7. Image resolution is poor.

22. Figures 8 and 9. These are multiple comparisons showing statistically significant differences between measures, but I don't believe any kind of correction (e.g. Bonferroni correction) was used. Some of the significant differences should probably be considered Type I errors. The x-labels are not consistent in both figures.

23. Line 380, Line 382, Line 394, and line 468. "adaptions" should be "adaptations".

24. Line 440. "singiner" should be "singers".

25. Line 441. "systematical" should be "systematic".

26. Lines 442-444. I don't understand what this sentence is trying to say.

27. Lines 445-446. I don't understand what this sentence is trying to say.

28. Line 471. "interindividually" I think should be deleted.

29. Line 472. "sex/fach/genre." What is "fach"?

30. Line 477. "patient" should be "patients".

31. I think could be made more clear at relevant places throughout the manuscript and in Figure 10, what "early" and "late" mean in this paper, i.e. relative to Figure 1.

Reviewer #3: The purpose of this study is to describe differences in kinematic movements related to respiratory physiology as singers produce sustained-pitch notes and rapidly change these notes to a higher or lower octave (termed a pitch jump). Measures of respiratory kinematics were obtained from dynamic magnetic resonance imaging (MRI) data at 3 frames per second. Measures were primarily derived from the position of the diaphragm relative to the apex of the lung as the singers dynamically changed pitch in an upward or downward direction. Recordings from an acoustic microphone and electroglottograph were also made during the MRI scans. In a separate session in a sound treated room, the singers produced p-vowel syllables at similar pitches as the MRI session to measure intraoral pressure and estimate the subglottal pressure produced during the sustained notes. The main result appears to be the finding that the diaphragm position (particularly in the posterior region) monotonically decreases during the sustained notes and octave jumps, except for the jump from the highest octave to the middle octave. In that context, the diaphragm temporarily reverses direction (as if during inspiration) and then continues to elevate as usual.

I found the paper to be very challenging to read. The data appear to be collected with appropriate methodologies; however, the motivation, analysis, and presentation of the results need significant improvement.

TITLE

I do not believe that the methodology addresses the study of respiratory regulation of subglottal pressure because these two measures were estimated in separate sessions. And the data presented by the investigators is on respiratory kinematics given octave pitch jumps.

ABSTRACT

"During sustained phonation, singers´ diaphragms were elevated and the rip cage diameter was reduced."

This sentence appears to state the diaphragm was eleveated throughout the sustained phonation. What I believe the authors mean is that the diaphragm continues to elevate in a monotonic manner throughout the sustained phonation. This particular sentence illustrates how the rest of the paper reads and is in need of improvement so the descriptions of data and results are clear.

"rip cage" should be changed to "rib cage" throughout.

"The magnitude of the movement correlated with the amount of subglottic pressure difference."

What is the evidence supporting this statement? I did not see any correlation statistic between diaphragm kinematics and subglottal pressure. Fig. 7 simply displays these variables but I do not see any reference to this highlighted result in the manuscript.

INTRODUCTION

"To the author’s knowledge only one attempt has been made using an imaging technique (ultrasound), but this study was limited by the low penetration depth of ultrasound into lung tissue (15)."

Even though the cited study had limitations, if it is the only other study investigating parameters of interest, then please much more should be discussed about this reference to put the current work in context. What did the prior work find and how does that lead into the current work?

"In a pilot study (22), the movements of the breathing apparatus of 6 professional singers during sustained phonation with a constant ƒo and SPL were analyzed. Here, a very sophisticated movement pattern was observed, which indicated a differentiated control of different parts of the DPH and RC."

Please describe more details about the pilot study. These types of statements are good for the introduction but require much more explanation. And please define what is meant by "sophisticated movement pattern."

The motivation behind the study appears to be in the following paragraph:

"Singing is seldom reduced to phonation of a single pitch but characterized by pitch jumps of different magnitude and direction. How different parts of the breathing system move with sudden pitch changes, has not previously been investigated. Thus, the aim of this study was to evaluate the movements of the breathing apparatus during phonation of pitch jumps."

However, the motivation that no one else has studied the phenomenon is not adequate. Why do the authors think this study is needed? How can one use the results?

The hypotheses are inadequate. The current hypotheses are too generic with statements like "respiratory movements differ" and "regulation of pitch jumps would not affect all parts of the respiratory system equally." Please be more specific as to which direction changes are expected and why.

METHODS

"At the time of the recording, none of the participants complained of any vocal symptoms or suffered from pulmonary disease (which was confirmed by the VC and FEV1 values in table 1)."

This statement mentions subject status at the time of recording. In addition, did any subject have a history of vocal complaints, voice disorders, or respiratory pathologies?

Section 2.5 MR Image Analysis

This section on MRI analysis is an important one that defines the primary respiratory kinematic measures of the study. More explanation and detail are necessary than the two sentences currently in this section. Were points on Figure 2 marked manually or with a computer algorithm? If manually, please describe the method used. Who did the ratings? What was their expertise? How many raters marked the images? Multiple raters should be used so measures of reliability can be reported. If algorithmic, please describe the algorithm.

The term "elevation" of the diaphragm is referred to in the Abstract and Discussion. Please explicitly define how diaphragm elevation was computed.

For example, is the DPHhighestP distance what the authors use to define elevation or other measure(s)? Also, "rib cage diameter" is referred to often. Is the rib cage diameter estimated using the apD_Thorax measure or other?

In my opinion, there are too many abbreviations. Please consider spelling out all terms for clarity for the reader. Please go through the manuscript and make sure terms are used consistently and spelled correctly. For example, new abbreviations were used that were not defined in Methods (DPHmed).

Please put scale bars and orientation directions (anterior, superior, etc.) on Figure 2 and 3.

"EGG-MRI signal": This signal is undefined. I was initially confused by what this term referred to. Do the authors just mean the EGG signal (recorded during MRI data collection)?

Section 2.5.2

There is an equivalence stated: "t_-4to+4 = m_1 - m_8". This is not an equivalence. Time does not equal distance. And the notation is very loose with a dash (minus sign?) between m_1 and m_8 and "to" between -4 and +4.

I highly recommend using the time notation defined (t_n) as x-axis labels in graphs in Fig 6 and 10 so it is clear when jump happens (t0). m1 to m8 are the mean gradients themselves that should be labeled on the y-axes. Fig 7 is labeled correctly.

Anorm is not used again in the manuscript. Is Anorm the y-axis in Fig 7?

Section 2.6

As mentioned in the abstract comments, the statement in the abstract, ""The magnitude of the movement correlated with the amount of subglottic pressure difference," is not supported in the current manuscript. The statistical analysis section does not mention a correlation between respiratory kinematics and subglottal pressure. Please address.

Fig 4 subglottal pressure plots: please correct the units on the y-axis to cm H2O.

Results

In general, while it is appreciated that figure captions include a statement of the result, please revise all caption to better describe/define what each graph, etc., is showing.

Section 3.2.1

Fig 6 is another example of wording that needs to be addressed. The text mention tjump, but there is not tjump in Fig 6. As mentioned above, the horizontal axis of Fig 4 should defined in terms of t_n (time).

Fig 6 and 7 display mean curves. Please add standard deviation or confidence intervals to provide the reader with an idea of variance across subjects.

Fig 8: Graph needs to be corrected for correct spelling of term apDthorax. The inconsistencies of terms across the manuscript make it very challenging to understand the study. Where is the DPHhighestP measure here? DPHmed here is not defined in Methods.

Fig 10: Please define "high down," "low down," "high up," and "low up" that is in the legend. These terms are not used anywhere else in the manuscript. Please only use terms in the Results section that have already been defined earlier in the manuscript.

"adaption" -> "adaptation"?

6. PLOS authors have the option to publish the peer review history of their article (what does this mean?). If published, this will include your full peer review and any attached files.

Reviewer #1: No

Reviewer #2: No

Reviewer #3: No

---

## [Author Response · Author response to Decision Letter 0]

16 Jan 2020

We thank the reviewers for their constructive critique and the helpful comments. We are trying to answer all reviewer questions individually in the rebuttal letter.

---

## [Decision Letter · Decision Letter 1]

27 Feb 2020

PONE-D-19-28236R1

Respiratory kinematics and regulation of subglottic pressure for phonation of pitch jumps – a real-time MRI study

PLOS ONE

Dear Dr Traser,

Thank you for submitting your manuscript to PLOS ONE. After careful consideration, we feel that it has merit but does not fully meet PLOS ONE’s publication criteria as it currently stands. Therefore, we invite you to submit a revised version of the manuscript that addresses the points raised during the review process.

We would appreciate receiving your revised manuscript by March 21. To enhance the reproducibility of your results, we recommend that if applicable you deposit your laboratory protocols in protocols.io, where a protocol can be assigned its own identifier (DOI) such that it can be cited independently in the future. For instructions see: http://journals.plos.org/plosone/s/submission-guidelines#loc-laboratory-protocols

We look forward to receiving your revised manuscript.

Kind regards,

Michael Döllinger, Ph.D.

Academic Editor

PLOS ONE

Journal Requirements:

Additional Editor Comments (if provided):

After reading the manuscrupt myself, I agree that the quality of the manuscript has been improved. Also the Topic and study is of hoigh interest. However, I share the concerns raised by the Reviewers that have to be addressed before acceptance, especially:

improve english, explain and justify performed statistics, justify why the authors switch between max and mean values.

Reviewers' comments:

Reviewer's Responses to Questions

**Comments to the Author**

1. If the authors have adequately addressed your comments raised in a previous round of review and you feel that this manuscript is now acceptable for publication, you may indicate that here to bypass the “Comments to the Author” section, enter your conflict of interest statement in the “Confidential to Editor” section, and submit your "Accept" recommendation.

Reviewer #1: (No Response)

Reviewer #2: (No Response)

2. Is the manuscript technically sound, and do the data support the conclusions?

Reviewer #1: Yes

Reviewer #2: Yes

3. Has the statistical analysis been performed appropriately and rigorously? 

Reviewer #1: No

Reviewer #2: Yes

4. Have the authors made all data underlying the findings in their manuscript fully available?

Reviewer #1: Yes

Reviewer #2: Yes

5. Is the manuscript presented in an intelligible fashion and written in standard English?

Reviewer #1: No

Reviewer #2: Yes

6. Review Comments to the Author

Reviewer #1: The authors have made considerable revisions and clarifications to the original manuscript. However, while I continue to find the premise, rationale, and data presented in the manuscript to be a valuable contribution to the field my concerns about the validity of the statistical analysis and the clarity of presentation in the manuscript have not been adequately addressed. I therefore cannot recommend this manuscript for publication in its present form.

Major concerns

1) The author's have not suitably addressed my previous concern that they sometimes analyse maximum measurements and sometimes mean measurements. Section 2.6 attempts to justify this practice, but the meaning of this section is not clear. A direct reading of this section suggests to me that the authors tested for differences between mean and maximum measurements, which would not be a very meaningful thing to do. I suspect that what they have done is to test which measurement gave the largest effects, but this practice is circular and will lead analyses to overestimate effects throughout the paper. The authors need to either provide an a-priori justification for their choices (which it is too late to do for these data), or report all tests and correct appropriately.

2) It is not clear to me that including subject (singer) as a covariate in the ANOVA's is sufficient to solve the problem of non-independence. Repeated measures ANOVA already removes variance between participants to deal with the fact that participants contribute data to each cell of the design. This does not address the problem that in some of the analyses reported, singers contribute multiple data points to the same cell of the design, which as far as I can tell remains a problem in the present manuscript. It should also be point out that the covariate of gender is redundant with the covariate of singer since any variation due to gender will already be accounted for. If it were very important to the authors, singer could be nested with gender do address that question, but I suspect that it would just be a digression.

3) In Section 3.2.1 All measurements are combined in a single test. This may not be the most meaningful way of testing hypothesis A. A value that is smaller than typical for one distance measure may be larger than typical for another. Likewise the various movement locations will have different ranges of values, which I suspect has left a heteroscedasticity problem. The authors should scale the movement directions if they are being combined. Perhaps some kind of multivariate analysis would be more appropriate. A simpler way of testing this hypothesis would be to analyse lung volume (as estimate by the area within the lung in 2d images).

4) The addition of the derived data sheets to the submission is appreciated. However, the authors should also consider including the raw MRI data. If including the raw audio data would not be consistent with protecting the privacy of participant's then by all means do not include it. Why not provide derived measures from the audio recordings (f0, SPL, etc.) which would then make the raw MRI data useable? I hope that the authors will consider this approach moving forward, as these data would be a useful asset to the field.

5) Despite many recommendations from all reviewers to improve the writing of the manuscript, there are still a large number of grammatical and spelling errors. In many cases this leads to low intelligibility.

Minor Concerns

L111 - hypothese

L125 - hypothized

L125-126 - replace with respectively with i.e.

L282 - pluralise cents, here and throughout

L309 - outliers

L355 - Tukey's-HSD, here and throughout

Many other typographical errors throughout

Reviewer #2: The authors have done a good job addressing my comments.

In spite of an acknowledgement for help from a native speaker, there are many grammatical mistakes throughout the revised text, and these will need to be corrected.

I would recommend the authors to consider referring to their MRI method as "dynamic MRI" rather than "real-time MRI", especially in the title, because "real-time MRI" sounds like rtMRI, which this study does not employ.

7. PLOS authors have the option to publish the peer review history of their article (what does this mean?). If published, this will include your full peer review and any attached files.

Reviewer #1: No

Reviewer #2: No

---

## [Author Response · Author response to Decision Letter 1]

17 Apr 2020

Response to the Reviewers

Editor Comments 

I share the concerns raised by the Reviewers that have to be addressed before acceptance, especially: 

• improve english, 

• explain and justify performed statistics, 

• justify why the authors switch between max and mean values.

We have answered these questions carefully, and we hope that the manuscript is now suitable for publication.

Reviewer #1 

The authors have made considerable revisions and clarifications to the original manuscript. However, while I continue to find the premise, rationale, and data presented in the manuscript to be a valuable contribution to the field my concerns about the validity of the statistical analysis and the clarity of presentation in the manuscript have not been adequately addressed. I therefore cannot recommend this manuscript for publication in its present form.

Major concerns

1) The author's have not suitably addressed my previous concern that they sometimes analyse maximum measurements and sometimes mean measurements. Section 2.6 attempts to justify this practice, but the meaning of this section is not clear. A direct reading of this section suggests to me that the authors tested for differences between mean and maximum measurements, which would not be a very meaningful thing to do. I suspect that what they have done is to test which measurement gave the largest effects, but this practice is circular and will lead analyses to overestimate effects throughout the paper. The authors need to either provide an a-priori justification for their choices (which it is too late to do for these data), or report all tests and correct appropriately.

We thank the reviewer for this comment as it shows that our analysis approach was not clearly explained in the manuscript. We augmented the statistic section in the method part, and the result section:

o Differences in movement pattern occur between upwards and downwards jumps. To test whether these differences are depend on the part of the respiratory system (i.e., the anatomical location) two different measures were extracted from the movement curves that optimally describe the characteristics of the movement: 

o For downwards jumps (which present with a short inversion of the movement direction) the “maximal gradient” (mmax) was extracted. This value represents the maximum inversion of the movement direction. As the maximum value is not meaningful for upwards jumps which are characterized by a steady movement, here the mean gradient (mmean) during the jump was used to evaluate whether different location presents a difference in steepness of the gradient over the whole jump window. Both values were statistically tested with an univariate ANOVA. Then, statistically significant differences were further analysed with a Turkey’s HSD post-hoc test. 

o The following flow chart illustrates the course of the statistical analysis, asso¬ciated hypotheses, and figures. 

The choice of maximum vs. mean values is clearly derived from the nature of the motion patterns. It is physically not meaningful to calculate maximum gradients for upward jumps, and/or mean gradients for downward jumps. Therefore, we decided again not to include them in the analysis, as they would not be interpretable. 

2) It is not clear to me that including subject (singer) as a covariate in the ANOVA's is sufficient to solve the problem of non-independence. Repeated measures ANOVA already removes variance between participants to deal with the fact that participants contribute data to each cell of the design. This does not address the problem that in some of the analyses reported, singers contribute multiple data points to the same cell of the design, which as far as I can tell remains a problem in the present manuscript. It should also be point out that the covariate of gender is redundant with the covariate of singer since any variation due to gender will already be accounted for. If it were very important to the authors, singer could be nested with gender do address that question, but I suspect that it would just be a digression.

After intensive statistical consultation we conclude that the chosen statistical measure is the best possible option because of the following reasons: 

o We agree with the reviewer, that the best way for a statistical analyzation of the presented data would be to consider only the same jump at each location of each subject differently. This would lead to n=7 which would not allow any statistic at all as the number is too small. 

o Including more subjects would be a good solution but professionally trained singers are very rare who are willing to take part in such a demanding study (singing pitch jumps in an MRI machine). 

o Another possibility would be to exclude any kind of statistic and simply visually describe the data.

o The method we chose is to pool the data and analyse it with a repeated measures ANOVAs that compares mean values across all variables which are based on repeated observations while controlling for confounding variables (like different subjects, locations and gender) using covariates

In the first version of the manuscript we only included the different subjects and gender as covariates but after the requests of the reviewer we additionally added the different locations as covariates. This changed the values of the statistical parameters, but not their significance levels. 

To make our approach (with all its limitations) more transparent a critical discussion of this part was included in the discussion section:

“Additionally, limitations of the statistic evaluation of the presented data must be mentioned. Due to the limited availability of professionally trained singers who are willing to take part in an MRI study and sing pitch jumps in a narrow MR scanner, only 7 subjects were included in this study. Whilst ideally each jump and each location would be analysed individually, a sample size of n=7 did not provide enough data for meaningful statistical analysis in this way. Therefore, our statistical approach included all data in repeated measures ANOVAs with covariates included to test for the effect of confounding variables (like different subjects, locations and gender). This approach was chosen as it allows certain statistical evaluation of the presented data, but the problem of non-independence and heteroscedasticity cannot be solved completely.”

3) In Section 3.2.1 All measurements are combined in a single test. This may not be the most meaningful way of testing hypothesis A. A value that is smaller than typical for one distance measure may be larger than typical for another. Likewise the various movement locations will have different ranges of values, which I suspect has left a heteroscedasticity problem. The authors should scale the movement directions if they are being combined. Perhaps some kind of multivariate analysis would be more appropriate. A simpler way of testing this hypothesis would be to analyse lung volume (as estimate by the area within the lung in 2d images).

We agree with the reviewer, that a heteroscedasticity problem cannot be finally excluded the presented data with the chosen statistic approach – to clarify this limitation to the reader we introduced a limitation section in the discussion section (see answer to point 2)

We do not agree with the reviewer, that quantification of lung volume from 2D areas would solve the statistical problem. The advantage of our methodology is that we can make statements on how the individual parts of the respiratory system are regulated in professional singers. In our opinion, this is more valuable then measurements of lung volume which can be acquired more easily by other techniques. 

4) The addition of the derived data sheets to the submission is appreciated. However, the authors should also consider including the raw MRI data. If including the raw audio data would not be consistent with protecting the privacy of participant's then by all means do not include it. Why not provide derived measures from the audio recordings (f0, SPL, etc.) which would then make the raw MRI data useable? I hope that the authors will consider this approach moving forward, as these data would be a useful asset to the field.

We made all measured data available for the reader – as supplemental data sheet and as individual movement curves in the supplement figures. The reader can therefor comprehend each individual movement, in each singer and each task. 

Additional, to show the quality of the MRI images, an example movie is enclosed, for which we have the permission to be published by the participant. 

We do not have the permission by all singers to publish their MR images as a movie even without sound.

For future studies we will seek the consent of future subjects for a publication of all images (but this comes at the risk that even less subjects are willing to take part of such a study). 

5) Despite many recommendations from all reviewers to improve the writing of the manuscript, there are still a large number of grammatical and spelling errors. In many cases this leads to low intelligibility.

The manuscript was checked by a native speaker with special attention on grammar and spelling

Minor Concerns

L111 – hypothese & L125 – hypothized � we changed to “the hypothesis”/ “the hypotheses” and “to hypothesise” all over the manuscript

L125-126 - replace with respectively with i.e.

L282 - pluralise cents, here and throughout

L309 - outliers

L355 - Tukey's-HSD, here and throughout

Many other typographical errors throughout � corrected

Reviewer #2

The authors have done a good job addressing my comments.

In spite of an acknowledgement for help from a native speaker, there are many grammatical mistakes throughout the revised text, and these will need to be corrected.

the manuscript was again checked with special attention on grammar and spelling

I would recommend the authors to consider referring to their MRI method as "dynamic MRI" rather than "real-time MRI", especially in the title, because "real-time MRI" sounds like rtMRI, which this study does not employ.

We thank the reviewer for this comment and changed the title as recommended to dynamic MR

---

## [Decision Letter · Decision Letter 2]

21 Oct 2020

Respiratory kinematics and the regulation of subglottic pressure for phonation of pitch jumps – a dynamic MRI study

PONE-D-19-28236R2

Dear Dr. Traser,

Thank you for submitting your manuscript to PLOS ONE. After careful consideration, we feel that it has merit but does not fully meet PLOS ONE's publication criteria as it currently stands. Therefore, we invite you to submit a revised version of the manuscript that addresses the points raised during the review process

We look forward to receiving your revised manuscript.

With kind regards,

Michelle Ciucci, Ph.D.

Academic Editor

PLOS ONE

Additional Editor Comments:

It appears that reviewer 1 takes a hard line on assumptions for statistical testing using analysis of variance. The authors did not do tests to ensure that the data meet the assumptions for ANOVA. As such, we don't know if the data are normally distributed  or if the variances are equal. ANOVA is pretty robust to violations of these assumptions. I think the best course of action is for the authors simply run these tests (Shapiro-Wilk for normal distribution and Levene's test for equal variance). Then, there are some decisions to make. If they reveal normal distribution and equal variance, then report that. If they do not, there are ways to transform the data. My guess is that if the authors do this and repeat the ANOVA, they will likely have the same results, or perhaps even more robust results. The alternative is to just state that the data failed these tests but ANOVA is pretty robust against these. The authors are welcome to contact me with any questions.

Reviewers' comments:

Reviewer's Responses to Questions

**Comments to the Author**

1. If the authors have adequately addressed your comments raised in a previous round of review and you feel that this manuscript is now acceptable for publication, you may indicate that here to bypass the “Comments to the Author” section, enter your conflict of interest statement in the “Confidential to Editor” section, and submit your "Accept" recommendation.

Reviewer #1: (No Response)

Reviewer #2: All comments have been addressed

2. Is the manuscript technically sound, and do the data support the conclusions?

Reviewer #1: Yes

Reviewer #2: Yes

3. Has the statistical analysis been performed appropriately and rigorously? 

Reviewer #1: No

Reviewer #2: Yes

4. Have the authors made all data underlying the findings in their manuscript fully available?

Reviewer #1: Yes

Reviewer #2: Yes

5. Is the manuscript presented in an intelligible fashion and written in standard English?

Reviewer #1: No

Reviewer #2: Yes

6. Review Comments to the Author

Reviewer #1: The approach to statistical analysis in this manuscript is not valid at present and the authors have been unable to find a suitable alternative. In particular, the analysis violates two of the core assumptions of the parametric statistical tests that are reported, namely 1) independence of observations and 2) homogeneity of variance. Either of these violations alone would render these analyses invalid.

While I recognise the potential value of this experiment, I do not see that this manuscript realises that potential and I cannot recommend it for publication.

Reviewer #2: I am satisfied with this revision and with the authors' replies to all review critiques.

7. PLOS authors have the option to publish the peer review history of their article (what does this mean?). If published, this will include your full peer review and any attached files.

Reviewer #1: No

Reviewer #2: No

---

## [Author Response · Author response to Decision Letter 2]

11 Nov 2020

Dear Dr. Ciucci, 

Thank you very much for picking up our manuscript as academic editor! 

As you suggested we again worked on the optimization of the statistical evaluation of the data. We thank you very much for your recommendations. They were included in the following way:

As you suggested we extended our statistical testing for the requirements of the repeated measures ANOVA of our curve gradients (m1-8): We used the Shapiro-Wilk test for normal distribution of the data. Probably due to the few subjects the data were not normally distributed, (p < .001). But, as you also already stated, the repeated measures ANOVA is believed to be very robust against normality violations and this requirement is therefore seen as the weakest one. That was demonstrated by Berkovits, I., Hancock, G. R., & Nevitt, J. (2000). Bootstrap Resampling Approaches for Repeated Measure Designs: Relative Robustness to Sphericity and Normality Violations. Educational and Psychological Measurement, 60(6), 877–892. and Vasey, M. W., & Thayer, J. F. (1987). The Continuing Problem of False Positives in Repeated Measures ANOVA in Psychophysiology: A Multivariate Solution. Psychophysiology, 24(4), 479–486.

In the next step, Mauchly's test was used to evaluate whether the sphericity assumption has been violated and was significant with p< .001. As ɛ >.75, the Greenhouse–Geisser adjustment was then used to correct for violations of sphericity. As the repeated measures ANOVA computes a one-way repeated measures ANOVA for the trimmed means, the homoscedasticity assumption is not required (Wilcox, R. (2012). Introduction to Robust Estimation and Hypothesis Testing (3rd ed.). Elsevier).

Thus in our opinion requirements are given to calculate the repeated measures ANOVA. These calculation are now included in the manuscript.

Still, we principally agree with Reviewer 1 that the few participants limit the approach to statistical analysis! However, as professional singers are a very special group of participants the number could not be raised. Reviewer 1 states correctly that our approach violates the independence of the observations. That clearly is a limitation, but also in other fields, like psychology, is common to analyze data from re-tests as independent values when inclusion of more subjects is impossible. This limitation was clearly mentioned in the limitation section of the discussion. 

The presented study analyzed professional singers’ behavior of the respiratory system during pitch jumps by direct visualization. The used technique enabled us to observe the diaphragm contraction during pitch jumps downwards for the first time. Nevertheless, the study design also revealed that this phenomenon is related to different requirements like the amount of subglottic pressure difference. That complicates the evaluation by only visual analyzation. Thus, in our opinion, the statistical evaluation helps the reader to follow the evaluation related to our hypotheses. Even if the statistic approach is limited and results are only of minor indicative value they are helpful to get a more distinct idea of which effect should be closer analysed in a bigger cohort in the future. In our opinion, it is just very important to point that clearly out to the reader, which we now did in the limitation section of the discussion. Thus, we really hope that the originality of the study will overcome the statistical limitation caused by the limited access of professional singers. The two other reviewers seem to be in agreement with that and thus we now hope for a positive outcome of this very long review process. 

Best regards, 

Louisa Traser

---

## [Decision Letter · Decision Letter 3]

24 Nov 2020

PONE-D-19-28236R3

Respiratory kinematics and the regulation of subglottic pressure for phonation of pitch jumps – a dynamic MRI study

PLOS ONE

Dear Dr. Traser,

Thank you for your patience as we handled transitioning editors and such. An expert in both signing and physiology has reviewed this manuscript and they have some minor suggestions. The system does not let me accept a manuscript and still require minor changes. So, first CONGRATULATIONS. Once you make these minor changes, please submit and I will accept for publication immediately without the need for re-review. Below is format letter. Thank you again for your persistence. This is an important study.

Thank you for submitting your manuscript to PLOS ONE. After careful consideration, we feel that it has merit but does not fully meet PLOS ONE’s publication criteria as it currently stands. Therefore, we invite you to submit a revised version of the manuscript that addresses the points raised during the review process.

We look forward to receiving your revised manuscript.

Kind regards,

Michelle Ciucci, PhD

Academic Editor

PLOS ONE

Reviewers' comments:

Reviewer's Responses to Questions

**Comments to the Author**

1. If the authors have adequately addressed your comments raised in a previous round of review and you feel that this manuscript is now acceptable for publication, you may indicate that here to bypass the “Comments to the Author” section, enter your conflict of interest statement in the “Confidential to Editor” section, and submit your "Accept" recommendation.

Reviewer #4: (No Response)

2. Is the manuscript technically sound, and do the data support the conclusions?

Reviewer #4: Partly

3. Has the statistical analysis been performed appropriately and rigorously? 

Reviewer #4: Yes

4. Have the authors made all data underlying the findings in their manuscript fully available?

Reviewer #4: Yes

5. Is the manuscript presented in an intelligible fashion and written in standard English?

Reviewer #4: No

6. Review Comments to the Author

Reviewer #4: This study uses novel methodology to simultaneously measure movement of multiple structures in the respiratory apparatus relative to pitch jumps during phonation in trained singers. The work should be commended for providing substantial insight into our understanding of how individual components of the respiratory subsystem of phonation contribute to fine biomechanical control and adaptation of voice production. All major limitations of this study have been addressed in previous reviews. Some minor limitations remain, and are listed below:

Abstract:

No comments

Introduction:

1. Lines 118-120 The sentence, "As during sustained phonation the movement range..." is unclear, perhaps because of the complex dependent clause early in the sentence. For ease of reading, the authors might consider the following re-write: "in our pilot study [18], the movement range of the posterior DPH was twice as large as the anterior DPH during sustained phonation. Thus, we additionally hypothesized that there would be differences in psub-adaptive movements of the anterior compared to posterior DPH during pitch jumps.”

2. "Pitch jumps" and "octave jumps" are used interchangeably throughout the manuscript. Based on Figure 2, all pitch jumps are octave jumps. The authors should choose either "pitch jumps" or "octave jumps" and use this throughout the manuscript. (ie, Table 1)

Methods

1. The authors do not state whether or not subjects intentionally maintained phonation in a given register (ie falsetto, modal, mixed). Vocal register is still not terribly well understood making discussion challenging, however, a comment on whether register cracks or breaks occurred, whether subjects intentionally maintained a “falsetto” register (which would likely change findings) or completed tasks as they would for performance, or some other mention of the subjects voice quality would facilitate repeatability and allow for a more-nuanced interpretation of findings.

2. Line 152: In table 2, subjects 2 and 3 are listed as belonging to Bunch/Chapman taxonomy "3.15b1." However, "3.15" does not exist in the taxonomy. Perhaps the authors meant "3.1b," indicating National/Big City Opera singers singing minor roles?

3. Line 161: The shape of the vocal tract will influence glottic configuration and subglottal pressure adaptation. For example, modifications to the vowel /a:/ are often implemented by classically trained singers to facilitate phonation in certain pitch ranges. The authors may wish to comment on whether they feel that subtle shifts in vowel (vocal tract) shape (both within and among subjects) might have influenced the measures of interest in the current study. This would be an appropriate addition to the discussion section.

4. Line 174: The authors should comment on the amount of time required for subjects to complete the pitch jumps. If this time was less than 1 second, it is possible that changes in DPH position occurred between samples taken from MRI (ie, might not be detected at 3fps). This would be a helpful addition to the discussion section

5. Lines 210-217: Measurement of Psub is taken during a task that is different from that performed during MRI. Because the task was not just different in terms of time (ie Psub was not measured during MRI) but also the task itself was different (/pa/ or /pi/ vs /a/), there may have been associated differences in MRI measures of interest. The authors should discuss this when interpreting any results that refer to Psub. In the discussion section.

6. Lines 210-217: Additionally, it is unclear when the /p/ portion of the /p/ occlusion task was performed. (ie, before, during or after the subject reached the target pitch jump?)

7. Line 281: In the sentence, "...are possibly influences differently..." it appears that "influences is supposed to be in past tense (ie, "...possibly influenced differently...")

8. Line 291: "Turkey's HSD" should be "Tukey's HSD"

Results:

No Comments

Discussion:

1. The authors may wish to discuss additional ways in which Psub might suddenly change (rather than inspiratory diaphragmatic movement). Ie, is there a way that this might be performed maladaptively at the level of the glottis (ie vocal fold tension) or in the vocal tract (pharyngeal constriction) that would be different between singers and normal controls? Pathological voices vs normal controls?

2. Lines 565-583: When "statistic" is used as an adjective, it is typically altered to become “statistical” (ie, "statistical evaluation," as opposed to "statistic evaluation.")

7. PLOS authors have the option to publish the peer review history of their article (what does this mean?). If published, this will include your full peer review and any attached files.

Reviewer #4: No

---

## [Author Response · Author response to Decision Letter 3]

8 Dec 2020

Answers to Reviewer #4: 

Reviewer #4: This study uses novel methodology to simultaneously measure movement of multiple structures in the respiratory apparatus relative to pitch jumps during phonation in trained singers. The work should be commended for providing substantial insight into our understanding of how individual components of the respiratory subsystem of phonation contribute to fine biomechanical control and adaptation of voice production. All major limitations of this study have been addressed in previous reviews. Some minor limitations remain, and are listed below:

Abstract:

No comments

Introduction:

1. Lines 118-120 The sentence, "As during sustained phonation the movement range..." is unclear, perhaps because of the complex dependent clause early in the sentence. For ease of reading, the authors might consider the following re-write: "in our pilot study [18], the movement range of the posterior DPH was twice as large as the anterior DPH during sustained phonation. Thus, we additionally hypothesized that there would be differences in psub-adaptive movements of the anterior compared to posterior DPH during pitch jumps.”

 Changed as suggested

2. "Pitch jumps" and "octave jumps" are used interchangeably throughout the manuscript. Based on Figure 2, all pitch jumps are octave jumps. The authors should choose either "pitch jumps" or "octave jumps" and use this throughout the manuscript. (ie, Table 1)

changed to pitch jumps throughout the manuscript

Methods

1. The authors do not state whether or not subjects intentionally maintained phonation in a given register (ie falsetto, modal, mixed). Vocal register is still not terribly well understood making discussion challenging, however, a comment on whether register cracks or breaks occurred, whether subjects intentionally maintained a “falsetto” register (which would likely change findings) or completed tasks as they would for performance, or some other mention of the subjects voice quality would facilitate repeatability and allow for a more-nuanced interpretation of findings.

We thank the reviewer for hat important suggestion. The subjects were asked to phonate in their western classically trained stage voice without register specification (we included that information in the method section). As you already stated the register terminology is unfortunately until now not standardized. The included professional singers succeeded in their aim to phonate the tasks without unintended register breaks. As the pitch jumps included the whole tessitura of the singers, certainly different registers conditions were used. That a singer used a natural falsetto is unlikely as it can be differentiated from stage voice by the lower sound pressure level and sound quality. The influence of register function on the presented data is discussed in a new paragraph in the discussion section. 

2. Line 152: In table 2, subjects 2 and 3 are listed as belonging to Bunch/Chapman taxonomy "3.15b1." However, "3.15" does not exist in the taxonomy. Perhaps the authors meant "3.1b," indicating National/Big City Opera singers singing minor roles?

Subjects 2 and 3 are professional choir singers in international touring vocal ensembles. 

We found that classification – 3.15b1 to be the closest to that

From: Bunch M, Chapman J. Taxonomy of singers used as subjects in 

scientific research. J Voice. 2000;14: 363–9.

3. Line 161: The shape of the vocal tract will influence glottic configuration and subglottal pressure adaptation. For example, modifications to the vowel /a:/ are often implemented by classically trained singers to facilitate phonation in certain pitch ranges. The authors may wish to comment on whether they feel that subtle shifts in vowel (vocal tract) shape (both within and among subjects) might have influenced the measures of interest in the current study. This would be an appropriate addition to the discussion section.

We agree! The vowel [a:] was chosen in all tasks to avoid the articulatory effects that can be expected when fundamental frequency exceeds the normal value of the first format. We discussed the question of interactions between articulation and respiratory movements in a new paragraph in the discussion section together with the question of registers.

4. Line 174: The authors should comment on the amount of time required for subjects to complete the pitch jumps. If this time was less than 1 second, it is possible that changes in DPH position occurred between samples taken from MRI (ie, might not be detected at 3fps). This would be a helpful addition to the discussion section

yes! We included the missing information in the method section and a also a paragraph in the limitation section of the discussion. 

5. Lines 210-217: Measurement of Psub is taken during a task that is different from that performed during MRI. Because the task was not just different in terms of time (ie Psub was not measured during MRI) but also the task itself was different (/pa/ or /pi/ vs /a/), there may have been associated differences in MRI measures of interest. The authors should discuss this when interpreting any results that refer to Psub. In the discussion section.

We thank the reviewer for this important comment and included the missing information (phonation of syllable repetition [pa:]) in the method section as well as in the discussion section. 

6. Lines 210-217: Additionally, it is unclear when the /p/ portion of the /p/ occlusion task was performed. (ie, before, during or after the subject reached the target pitch jump?)

The syllable [pa:] was repeated 3 times on each pitch, starting the new pitch with a [p:]. Each pitch was held for 3 seconds. 

7. Line 281: In the sentence, "...are possibly influences differently..." it appears that "influences is supposed to be in past tense (ie, "...possibly influenced differently...")

Corrected as suggested

8. Line 291: "Turkey's HSD" should be "Tukey's HSD"

Corrected as suggested

Results:

No Comments

Discussion:

1. The authors may wish to discuss additional ways in which Psub might suddenly change (rather than inspiratory diaphragmatic movement). Ie, is there a way that this might be performed maladaptively at the level of the glottis (ie vocal fold tension) or in the vocal tract (pharyngeal constriction) that would be different between singers and normal controls? Pathological voices vs normal controls?

We thank the reviewer for this interesting idea. We included a paragraph in the discussion section. As our primary outcome parameter is the DPH movement we discussed whether this adaptational movement of the DPH to reduce psub is economic. It could be speculated that a failure to reduce the psub by the breathing apparatus could be associated with singing out of tune. However, also glottal, or vocal tract adaptions affect psub. Malregulation of the respiratory system could therefore be related to the necessity of adaptations on glottal or vocal tract level.

2. Lines 565-583: When "statistic" is used as an adjective, it is typically altered to become “statistical” (ie, "statistical evaluation," as opposed to "statistic evaluation.")

Corrected as suggested

---

## [Editor Report · Decision Letter 4]

14 Dec 2020

Respiratory kinematics and the regulation of subglottic pressure for phonation of pitch jumps – a dynamic MRI study

PONE-D-19-28236R4

Dear Dr. Traser,

Congratulations! We’re pleased to inform you that your manuscript has been judged scientifically suitable for publication and will be formally accepted for publication once it meets all outstanding technical requirements.

Although this was a long process, I think the end result is a terrific paper on a topic that needs more scientific study. Best of luck to you.

Kind regards,

Michelle Ciucci, PhD

Academic Editor

PLOS ONE
---

## [Editor Report · Acceptance letter]

22 Dec 2020

PONE-D-19-28236R4 

Respiratory kinematics and the regulation of subglottic pressure for phonation of pitch jumps – a dynamic MRI study 

Dear Dr. Traser:

I'm pleased to inform you that your manuscript has been deemed suitable for publication in PLOS ONE. Congratulations! Your manuscript is now with our production department. 

Kind regards, 

on behalf of

Dr. Michelle Ciucci 

Academic Editor

PLOS ONE